# An Upper-branch Brewer Dobson Circulation index for attribution of stratospheric variability and improved ozone and temperature trend analysis

William T. Ball[1,2], Aleš Kuchař[3], Eugene V. Rozanov[1,2], Johannes Staehelin[1], Fiona Tummon[1], Anne K. Smith[4], Timofei Sukhodolov[1,2], Andrea Stenke[1], Laura Revell[1,5], Ancelin Coulon[1], Werner Schmutz[2], and Thomas Peter[1]

[1]Institute for Atmospheric and Climate Science, Swiss Federal Institute of Technology Zurich, Universitaetstrasse 16, CHN, CH-8092 Zurich, Switzerland
[2]Physikalisch-Meteorologisches Observatorium Davos World Radiation Centre, Dorfstrasse 33, 7260 Davos Dorf, Switzerland
[3]Department of Atmospheric Physics, Faculty of Mathematics and Physics, Charles University in Prague, V Holesovickach 2, 180 00 Prague 8, Czech Republic
[4]National Center for Atmospheric Research, Boulder, Colorado
[5]Bodeker Scientific, Alexandra, New Zealand

*Correspondence to:* W. T. Ball (william.ball@env.ethz.ch)

**Abstract.** We find that wintertime temperature anomalies near 4 hPa and 50°N/S are related, through dynamics, to anomalies in ozone and temperature, particularly in the tropical stratosphere, but also throughout the upper stratosphere and mesosphere. These mid-latitude anomalies occur on timescales of up to a month, and are related to changes in wave-forcing. A change in the meridional Brewer-Dobson circulation extends from the middle stratosphere into the mesosphere and forms a temperature-change quadrupole from equator to pole. We develop a dynamical index based on detrended, deseasonalised mid-latitude temperature. When employed in multiple linear regression, this index can account for up to 60% of the total variability of temperature, peaking at ∼5 hPa and dropping to zero at ∼50 hPa and ∼0.5 hPa, and increasing again into the mesosphere. Ozone similarly sees up to an additional 50% of variability accounted for, with a slightly higher maximum and strong altitude dependence, with zero improvement found at 10 hPa. Further, the uncertainty on all equatorial multiple-linear regression coefficients can be reduced by up to 35% and 20% in temperature and ozone, respectively, and so this index is an important tool for quantifying current and future ozone recovery.

## 1 Introduction

Trend analysis, typically using multiple linear regression (MLR), is a key approach to understand drivers of long-term changes in the stratosphere (e.g. WMO (1994), Soukharev and Hood (2006), Chiodo et al. (2014), Kuchar et al. (2015), Harris et al. (2015)). Ozone and temperature have received most attention, partly because they have the longest observational records. Temperature is important for understanding climate change, while quantifying changes in the ozone layer is necessary to estimate

the impacts of elevated, or reduced, ultraviolet (UV) radiation reaching the surface, especially following the implementation of the Montreal Protocol to reduce halogen-containing ozone depleting substances (ODSs).

Ozone and temperature variations in the stratosphere are directly modulated by changes in solar flux, particularly in the UV (see e.g. Haigh et al. (2010); Ball et al. (2016) and references therein). Ozone concentration also responds to changes in the Brewer-Dobson circulation (BDC), whereby air rises in the tropics, advects polewards either on a lower, shallow-(below $\sim$50 hPa) or an upper, deep-branch, and descends at mid-latitudes (less than $\sim$60°) or over the poles, respectively (Birner and Bönisch, 2011). The BDC is mainly driven by mid-latitude upward propagating planetary and gravity waves that break and impart momentum, acting like a paddle to drive the circulation (Haynes et al., 1991; Holton et al., 1995; Butchart, 2014). Wave forcing depends on the mean-state of the flow, and vice-versa (Charney and Drazin, 1961; Holton and Mass, 1976); changes in either affect ozone transport by a change in the speed of the BDC that leads to adiabatic heating, or cooling, and directly affects chemistry through temperature-dependent reaction rates (Chen et al., 2003; García-Herrera et al., 2006; Shepherd et al., 2007; Lima et al., 2012). As such, ozone and temperature have an inverse relationship in the equatorial stratosphere above 10 hPa, which in turn has a dependence on dynamics (Fusco and Salby, 1999; Mäder et al., 2007; Stolarski et al., 2012), although this is not always the case in the lower stratosphere (Zubov et al., 2013). Ultimately, then, dynamical perturbations at mid-to-high latitudes can directly influence the variability of ozone and temperature (Sridharan et al., 2012; Nath and Sridharan, 2015).

The stratospheric ozone layer has been damaged by the use of ODSs and following a ban through the 1987 Montreal Protocol (Solomon, 1999), levels of ODSs have declined since their peak in 1998 (Egorova et al., 2013; Chipperfield et al., 2015), although the peak may be earlier or later depending on the location of interest. However, the rate of ozone recovery is latitude dependent, with southern mid-to-high latitudes expected to recover from elevated ODSs (WMO, 2011). The increase in ozone at mid-latitudes comes partly from ODSs reductions, but also because the BDC is expected to accelerate (Garcia and Randel, 2008; Butchart and Scaife, 2001; Butchart, 2014), which will reduce the time for ozone depletion to occur and lead to faster transport of ozone from the equatorial region to higher latitudes. This in turn leads to a reduction of ozone over the equator and a prevention of a full recovery over the tropics. Thus, the recovery of ozone at mid-to-high latitudes can be understood as being partly due to less ozone destruction by lower ODS concentrations, and partly due to a faster redistribution of ozone-rich air from the tropics. Additionally, the cooling stratosphere will slow ozone depletion and further support the increase in ozone at mid-latitudes (WMO, 2014). However, estimates of decadal trends in ozone since 1998 have a high level of uncertainty (Harris et al., 2015) because various long-term datasets provide different pictures (Tummon et al., 2015), and we do not understand much of the stratospheric variability on short timescales. Anomalous, monthly variability, like that at the equator as shown in Figs. 7 and 8 in Shapiro et al. (2013), and which could be related to high latitude variability (e.g. Kuroda and Kodera (2001) and Hitchcock et al. (2013)) may simply be considered as noise in MLR trend estimates (and other regressors) where it is not accounted for, which increases the uncertainty.

In MLR analysis of the equatorial stratosphere, variability is usually described with at least six regressors that represent the solar cycle UV flux changes (e.g., with the F10.7cm radio flux), volcanic eruptions (stratospheric aerosol optical depth; SAOD), the El Nino Southern Oscillation (ENSO) surface temperature variations, two orthogonal modes of the dynamical

quasi-biennial oscillation (QBO), and the equivalent effective stratospheric chlorine (EESC), which describes the long-term influence of ODSs on ozone concentration and temperature. A green-house gas (GHG) proxy is sometimes also considered; or, alternatively to applying both GHG and EESC proxies, a linear (or piece-wise linear) trend is considered.

At higher latitudes, other proxies have been used to represent dynamical indices, e.g. in the northern hemisphere, the North Atlantic Oscillation (NAO) and Arctic Oscillation (AO), which are related to surface pressure changes, though their relation is less anti-correlated with stratospheric variability than, e.g., the tropopause pressure (Weiss et al., 2001). Trends in dynamically related quantities, such as horizontal advection and mass divergence, contribute to long-term changes in ozone (Wohltmann et al., 2007). For short timescales, Wohltmann et al. (2007) note that tropospheric pressure is a physical quantity directly responsible for changes in lower stratospheric temperatures, but that it is nevertheless inferior to stratospheric temperature when accounting for ozone and temperature variance in column ozone and temperature; this can also depend on the location of tropospheric blocking events (WMO, 2014). However, longer timescales render these proxies unreliable due to additional radiative effects. Ziemke et al. (1997) identified that the use of high-latitude temperature at 10 hPa in winter-spring months, together with 200 hPa temperature at mid-latitudes all year round, were most effective at reducing residuals, though Ziemke et al. (1997) limited their study to total column ozone.

In fact, several studies have identified proxies, such as temperature in the stratosphere, that can help improve MLR analysis (e.g., Ziemke et al. (1997); Appenzeller et al. (2000); Weiss et al. (2001); Mäder et al. (2007)). However, these have tended to be focused on total column ozone, the mid-to-lower stratosphere, or mid-latitude and polar regions, and thus most attention on dynamical variability remains associated with the lower branch of the BDC (e.g. Newman et al. (2001); Wohltmann et al. (2005); Brunner et al. (2006)). Further, studies of dynamical variability have also tended to focus on seasonal and inter-annual timescales, and thus any fluctuations on monthly or shorter timescales may be missed, underestimated, or driven by processes operating on different timescales.

There are differing conceptual approaches to improve regression models (i.e., see WMO (2011)): use of a statistical approach (e.g. Mäder et al. (2007)); or to use proxies that can be (at least partly) physically understood (e.g. Wohltmann et al. (2007)). Both approaches have their limitations and the physical mechanisms may not be fully understood in either case. As Wohltmann et al. (2007) points out, the use of (or lack thereof) unphysical or too many regressors could lead to systematic errors - through the attribution of correlated variables - that go unnoticed because error statistics do not change, or indeed, decrease. A third approach simply considers dynamical variability as noise that leads to enhanced uncertainties on trend analysis.

The identification of a correlation between two variables can be considered a first step in identifying the physical mechanism that underlies the causal link. A relationship between the proxies and processes that drive their variability need to be shown through additional information, it cannot be done simply through statistical means alone as it needs information from a physical understanding, either a priori, or following further investigation. Here, our aim is to find an index, or proxy, that represents rapid changes**, on timescales of a month or less,** in the upper branch of the BDC by investigating an identified association between temperature variation in the mid-latitude upper stratosphere and planetary wave-breaking. While temperature alone does not represent a complete picture of the physical driver of rapid BDC changes, we show that its variance is highly associated with wave-driving and that it can act as a proxy for such processes, especially where the exact physical drivers remain unresolved; the

use of standard dynamical proxies is, as we shall show, not enough to capture this variance. Chandra (1986) identified similar short-term dynamical variability that we identify in monthly data here, but he applied it to understand dynamical influences on upper stratospheric variability relevant to identifying 27-day solar irradiance variability - indeed showing that 27-day solar modulation was very difficult to identify due to large, rapid dynamical fluctuations - and did not extrapolate this information to

improving MLR analysis, as we aim to do here. The drawback to using temperature is that it mixes processes that might have different influences on ozone (Wohltmann et al., 2005). However, it is a simple and direct measure of dynamical changes, at least on monthly or shorter timescales, that relate to rapid dynamical adjustments within the stratosphere.

In summary, our aim here is to provide an index (section 4) to account for sporadic, noise-like stratospheric variability in monthly timeseries that represents rapid adjustments in the BDC and, therefore, better account for residual variance, improve

estimates of trends and regressor variability, and reduce their uncertainties (section 5). We do this using model, reanalysis and observational data (section 2) to identify a source for the short-term variability (section 3).

## 2   Data and models

### 2.1   Chemistry climate model in specified dynamics mode

To investigate temperature and ozone variability in the stratosphere and mesosphere at all latitudes, without data gaps, we

simulate historical ozone and temperature variations using the Chemistry Climate Model (CCM) SOlar Climate Ozone Links (SOCOL; version 3 (Stenke et al., 2013)) in specified dynamics mode, whereby the vorticity and divergence of the wind fields, temperature and the logarithm of surface pressure are 'nudged' using the ERA-Interim reanalysis (Dee et al., 2011) between 1983–2012 and up to 0.01 hPa; see Ball et al. (2016) for full nudging details. Note that we use the Stratospheric Processes and their Role in Climate (SPARC)/International Global Atmospheric Chemistry (IGAC) Chemistry Climate Model

Intercomparison (CCMI) boundary conditions and external forcings (Revell et al., 2015), except for the solar irradiance input, for which we use the SATIRE-S model (Krivova et al., 2003; Yeo et al., 2014). In the following we focus on temperature and ozone variables; the former is nudged, while the latter is simulated by the CCM SOCOL.

### 2.2   Observations

We verify that the nudged-model output fields ozone (not nudged) and temperature (nudged) agree with observations. For ozone

we use the Stratospheric Water and OzOne Satellite Homogenized (SWOOSH) ozone composite (Davis et al., 2016) for 215–0.2 hPa ($\sim$10–55 km) at all latitudes. For temperature, we compare the nudged-model output with independent measurements from the Sounding of the Atmosphere using Broadband Emission Radiometry (SABER) instrument (Russell et al., 1999) on the Thermosphere-Ionosphere-Mesosphere-Energetics and Dynamics (TIMED) satellite, spanning 2002–2015 and for 100 to 0.00001 hPa ($\sim$10–140 km) and latitudes out to $52°$.

For the MLR analysis (section 5) we additionally consider equatorial ozone from the Global OZone Chemistry And Related trace gas Data records for the Stratosphere (GOZCARDS; Froidevaux et al. (2015)), Solar Backscatter Ultraviolet Instrument

Merged Cohesive (SBUV-Mer.; Wild and Long (2016)), SBUV Merged Ozone Dataset (SBUV-MOD; Frith et al. (2014)) composites and temperature from the Stratospheric Sounding Unit observations (SSU; Zou et al. (2014)), and JRA-55 (Ebita et al., 2011) and MERRA (Rienecker et al., 2011) reanalyses. All observations are re-gridded onto the SOCOL model pressure levels and latitudes. We consider monthly mean zonally-averaged data.

## 3 Anomalous dynamical variability

### 3.1 Equatorial ozone and temperature variability

We define short-term, 'anomalous', variability here to be that occuring on monthly, or shorter, timescales. To identify this rapid variability, distinct from behaviour on seasonal and longer timescales, we remove all long-term variability by subtracting a timeseries that has been smoothed, with a 13-month running mean, and then deseasonalised, with monthly values, at each latitude and pressure. We apply this pre-processing to all variables described in sections 3 and 4. An example equatorial (20°S–20°N) ozone and temperature anomaly timeseries from the CCM SOCOL at 2.5 hPa is shown in Fig. 1. SWOOSH ozone from 1985 to 2012, and SABER temperature from 2002 to 2012 (Fig. 1), show similar anomalies to the model and have correlation coefficients ($r_c$) of 0.72 and 0.83 with the nudged model results, respectively; the model, therefore, reproduces observations well. The monthly temperature and ozone anomalies have a very strong relationship, especially between 0.1 and 6.3 hPa, with negative $r_c$ reaching -0.96 (Fig. 2) between 0.1 and 10 hPa, while being positive elsewhere.

To establish the coherency of the ozone-temperature relationship in the tropics, we identify 'extreme' anomalies (or 'events') as those at least at the 90th percentile from the mean in temperature and at less than the 10th percentile for ozone (and vice versa). We call 'low-T' events those that have low equatorial temperature at the same time as a high ozone concentration (blue lines at 2.5 hPa, Fig. 2), and 'high-T' for the opposite situation (red lines); the Low-T thresholds are -1.3 K for temperature and +2.4% for ozone, while High-T thresholds are +1.1 K and -2.2% (these are also given the upper plot of Fig. 1). We note that the ozone mixing ratio maximum in parts per million (ppm) is at ∼10 hPa. We use 2.5 hPa as a reference here, but other pressure levels at altitudes between 0.1 and 6 hPa give similar results. The majority of the events (45/60) occur in December-January-February (DJF; red/blue in Fig. 2) and June-July-August (JJA) (yellow/turquoise in Fig. 2). High-T and low-T months remain grouped above 10 hPa, but mix and lose coherence at altitudes below 10 hPa, implying that the events have a similar source at all altitudes above 10 hPa, but a different one below (i.e. $r_c$ is high at 25 and 40 hPa, but the events at 25 hPa are well-mixed). This indicates a likely transition between BDC branches and that the driver of variability is dynamical, which we confirm in the following.

### 3.2 Mid-latitude temperature variability

To identify and locate the source of the driver behind ozone and temperature anomalies shown and described in the previous section, in Fig. 3 we correlate the 2.5 hPa equatorial temperature low-T and high-T events with detrended and deseasonalised temperature at all latitudes and pressure-levels, for DJF and JJA months (Figs. 3a and b, respectively). A quadrupole-like

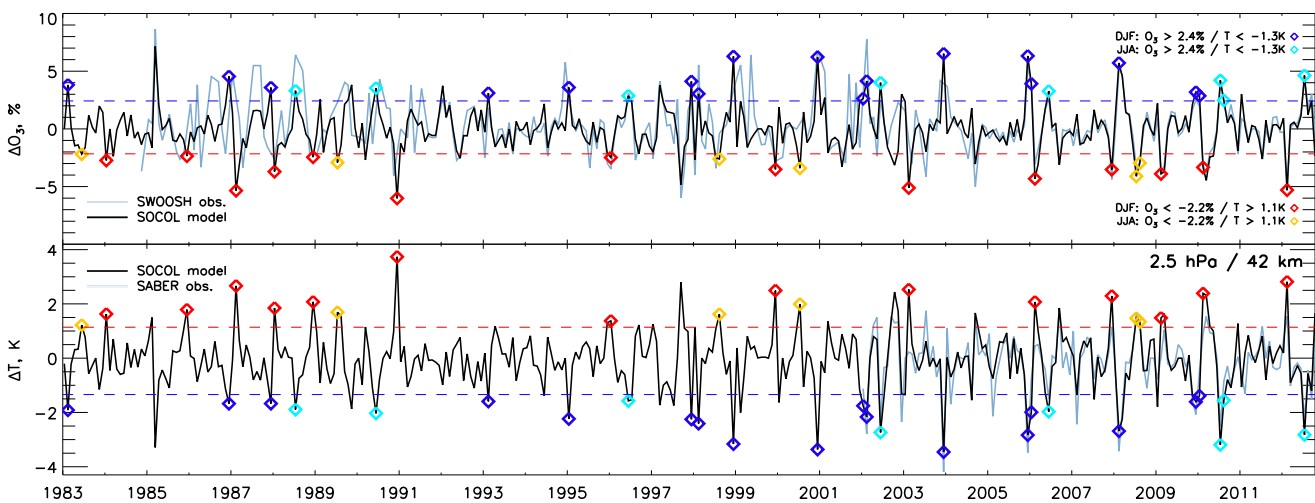

**Figure 1.** Monthly anomalies of equatorial (20°S–20°N) ozone (upper; %) and temperature (lower; degrees Kelvin) at 2.5 hPa, following the subtraction of 13-month box-car smoothing and monthly-deseasonalising from the CCM SOCOL model in specified dynamics mode. SWOOSH ozone composite timeseries and SABER temperature measurements are shown in light-blue in the upper and lower plots, respectively. The dashed blue and red horizontal lines are the thresholds shown in Fig. 2; thresholds for each coloured diamond are given on the right of the upper panel. June-July-August (JJA) anomalies exceeding the thresholds have orange (high-T) and turquoise (low-T) diamonds; December-January-February (DJF) anomalies are identified by red (high-T) and blue (low-T) diamonds.

structure emerges with positive correlations centred around 2.5 hPa at the equator and in the winter-polar mesosphere ($<0.8$ hPa), and negative correlations in the winter stratosphere at mid-to-high-latitudes and in the equatorial mesosphere. The inverse correlation in the stratosphere for DJF extreme months peaks at $\sim$52°N ($r_c$= -0.92); while JJA events peak at $\sim$43°S ($r_c$= -0.93). We find similar results when using other equatorial pressure levels near 2.5 hPa as a reference to calculate correlations.

5     Figures 3c–f show temperature composites for each event type: (c) DJF Low-T, (d) JJA Low-T, (e) DJF High-T and (f) JJA High-T; all show the same temperature-quadrupole structure as in Fig. 3a–b (signals at two and three standard deviations from zero are given as yellow and blue contours, respectively). Equatorial temperature anomalies ($\sim$2 K) are smaller than at high latitudes ($\sim$5 K or more). The maximum temperature response at mid-to-high latitudes does not always reside at the same location as the peak correlation. Although the statistics are less robust, since the period is shorter, the quadrupole structure is 10 also evident in SABER observations (Fig. 4). Thus, we can be confident that the nudged-model is giving a good representation of observations.

    The quadrupole structure is likely the result of (i) an acceleration of the BDC that adiabatically cools the equator during Low-T events as more air arrives at high-latitudes, and adiabatically heats there, and (ii) a deceleration of the BDC that adiabatically heats the equator during High-T events as less air arrives at high-latitudes leading to cooling there; both processes are associated 15 with changes in wave activity.

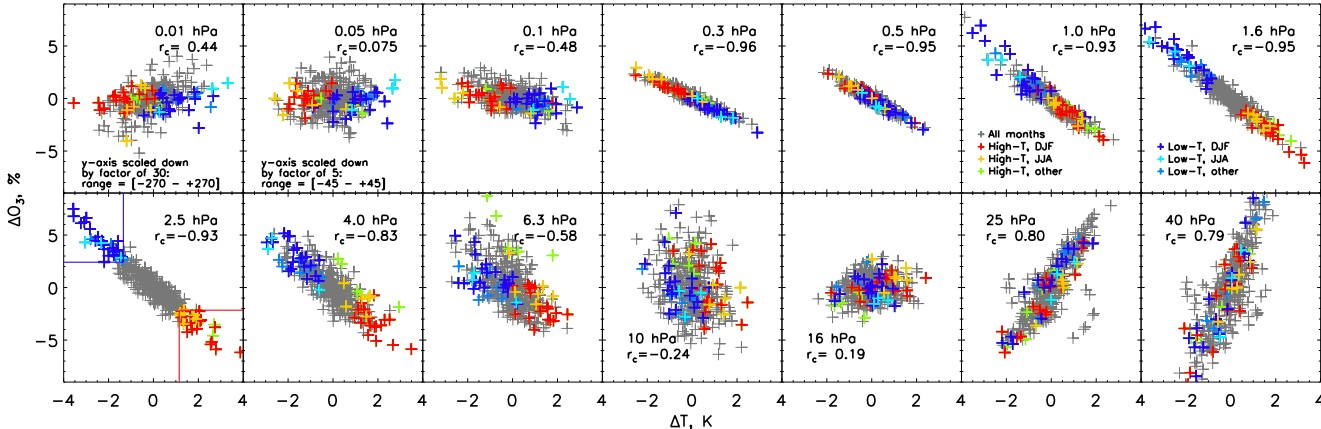

**Figure 2.** Regression of equatorial (20°N–20°S) ozone and temperature anomalies (following 13-month smoothing and monthly deseason-alising) from the CCM SOCOL model in specified dynamics mode for pressure levels 0.01 to 40 hPa (∼80 – 22 km). Grey crosses are for all other months in 1983/01–2012/10. Coloured crosses in each plot are determined at 2.5 hPa (lower-left, and plotted as diamonds in Fig. 1) by those within regions defined by the red (high-T events) and blue lines (low-T events); red crosses are for high-T events in December, January, and February (DJF), yellow for high-T events in June, July and August (JJA), and green for 'other' high-T events. Dark blue, turquoise and blue represent DJF, JJA and 'other' low-T months (see also legends in 1.0 and 1.6 hPa plots). Correlation coefficients are given for all crosses together. The y-scale has been decreased by a factor of 30 and 5 at 0.01 and 0.05 hPa, respectively, as indicated in the plots.

We show that the mid-latitude temperature, as well as the equatorial temperature and ozone anomalies are related to variations in wave activity using the Transformed Eulerian Mean streamfunction (TEMS; Figs. 5), a measure of the mass flux (positive values imply clockwise flow along contours, negative anti-clockwise) and the Eliassen-Palm Flux divergence (EPFD; Figs. 6), which is a measure of the resolved wave-induced forcing of the mean flow (positive values imply an acceleration of the zonal mean flow and a deceleration of the BDC, and negative values the opposite). **We used six-hourly model output to calculate the monthly means and use equations 3.5.1 and 3.5.3 from Andrews et al. (1987) to perform the calculations.** Using the events identified in Figs. 1 and 2, and used in Fig. 3, we find clear EPFD and TEMS anomalies centred near 55°, slightly poleward of the mid-to-high latitude peak correlations (Fig. 3a–b). As anomalies, they do not represent a reversal of meridional air flow, but a slowing or acceleration. When high-T anomalies occur the EPFD is positive, which implies zonal mean westerly winds have accelerated and the BDC has slowed, which is confirmed by the TEMS, indicating increased equatorward flow. This will have the exact effect found, of adiabatically heating the equatorial region and cooling the mid-to-high latitudes relative to the mean state. The opposite is the case for low-T anomalies. These results confirm that equatorial anomalies are dynamically driven and we suggest that it appears to be mainly related to, at the equator, a shift in the ozone maximum upwards during low-T events that then produces the anti-correlation seen in Fig. 2 above 10 hPa, and vice versa during high-T events. A further consequence of the circulation changes for ozone is that a temperature increase should lead to faster catalytic

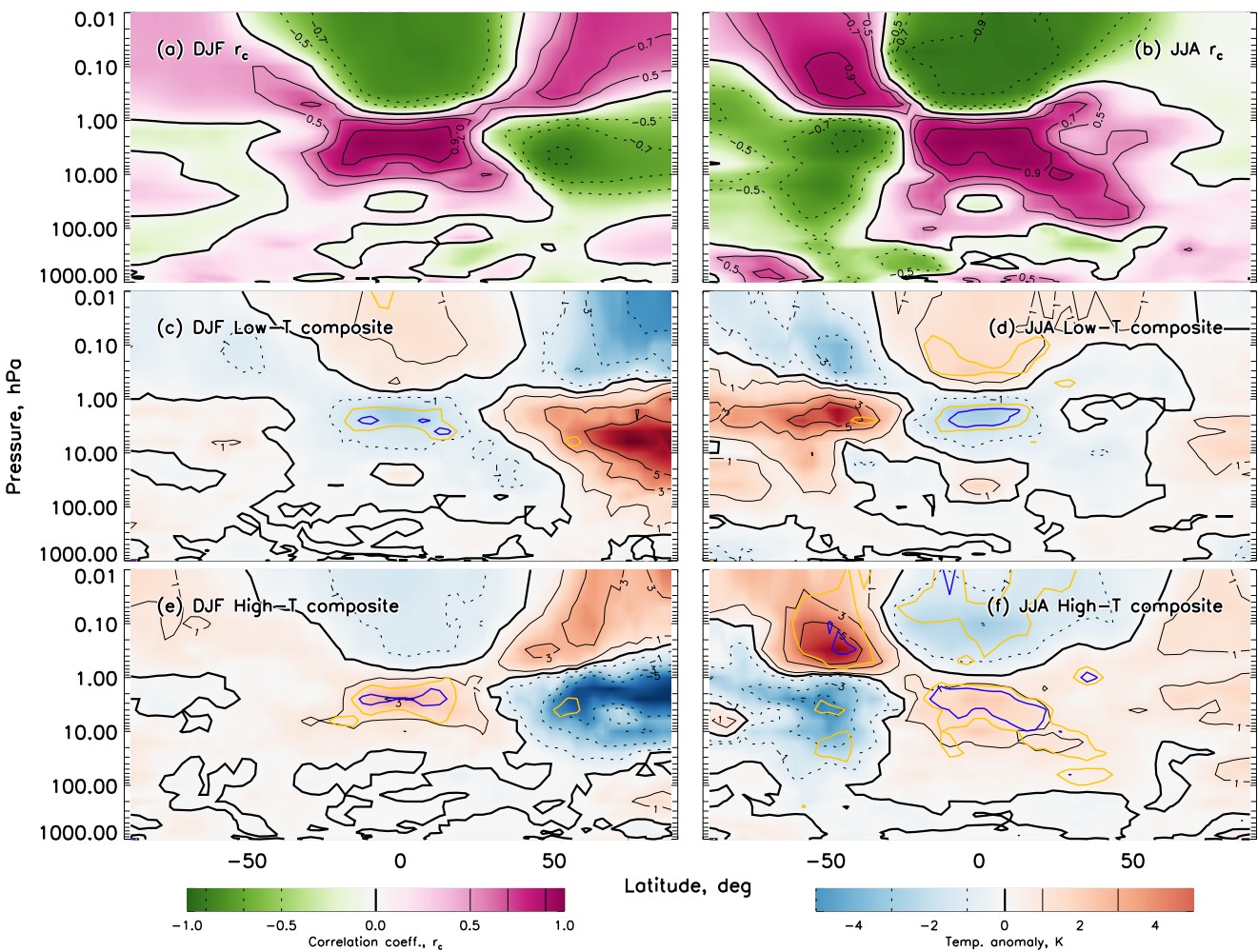

**Figure 3.** Correlation coefficient maps of zonal mean 20°N–20°S 2.5 hPa temperature anomalies from the SOCOL model with respect to latitude and altitude for all identified low- and high-T (a) DJF and (b) JJA events, as defined in Fig. 2. (**c–f**) Composite temperatures for (c) DJF low-T, (d) JJA low-T, (e) DJF high-T and (f) JJA high-T events. Dashed (solid) contours are negative (positive) with the bold line representing zero. Signals at the 2 and 3 standard deviations from zero are given as yellow and blue contours, respectively, in panels c–f.

destruction, and therefore a decrease of ozone, and vice versa for temperature decreases, though these effects seem to be less important than the rapid profile adjustment itself.

## 4 Upper-branch Brewer Dobson Circulation (UBDC) index

The link between anomalous mid-latitude temperature changes and equatorial temperature and ozone provides a way to account
5    for sporadic variability. When performing, e.g., an MLR analysis to understand variability in the stratosphere, such an index of

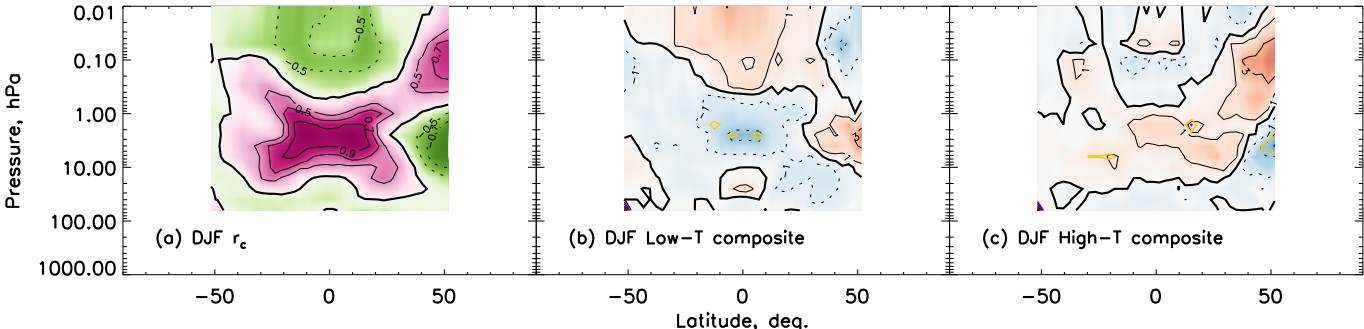

**Figure 4.** (**a**) SABER temperature data correlation coefficient map of zonal mean 20°N–20°S 2.5 hPa anomalies with all latitudes and altitudes for all low- and high-T DJF and events. Composite temperatures for DJF (**b**) low-T and (**c**) high-T events, as defined in Fig 2. Shading colours and black contours are the same as in Fig. 3 (dashed, negative; solid, positive; thick, zero). Signals at two standard deviations from zero are given as yellow contours in panels b and c.

monthly anomalies can account for a large proportion of variability previously unaccounted for, and drive down uncertainties on regressor estimates. We focus here on the equatorial region, but our results imply this index could be applied to other locations in the stratosphere and mesosphere.

Below, we describe how we construct an upper-branch Brewer Dobson circulation (UBDC) index based on detrended and deseasonalised temperature averaged over 43–49°S and 2.5–6.3 hPa for June–October, and averaged over 52–57°N and 4–10 hPa for November–May. Our index utilizes the output from the CCM SOCOL in specified dynamics mode, similar to ERA-Interim and observations, but such an index could be constructed in a similar way for any specific model.

### 4.1 Construction

To construct a useful upper-branch Brewer Dobson circulation (UBDC) index requires the identification of maximum correlation between the equator and each hemisphere separately, followed by a combination of information from these two regions. We have previously considered just the extreme events, but we now consider all monthly anomalies between 1983 and 2012. While wave-activity drives the temperature changes, it is not an easily observable quantity. Thus, temperature is a natural and simple quantity to build the index with. Additionally, we have found that the CCM SOCOL in free-running mode (i.e. without nudging) shows the same anomalous temperature-quadrupole structure as in Fig. 3 (not shown). Therefore, one can easily construct an index using model data to represent anomalous behaviour in the equatorial regions, and elsewhere where there is a quadrupole response.

We identify the maximum inverse temperature correlations at mid-latitudes in both DJF and JJA by varying the reference equatorial pressure-level. We find that averaging over the nine grid-cells centred on the mid-latitude peak improves the relationship with the equatorial region. Therefore, we construct the index with anomalous temperatures averaged over 43–49°S and 2.5–6.3 hPa in the southern hemisphere (SH) JJA months, and 52–57°N and 4–10 hPa in the northern hemisphere (NH) DJF months.

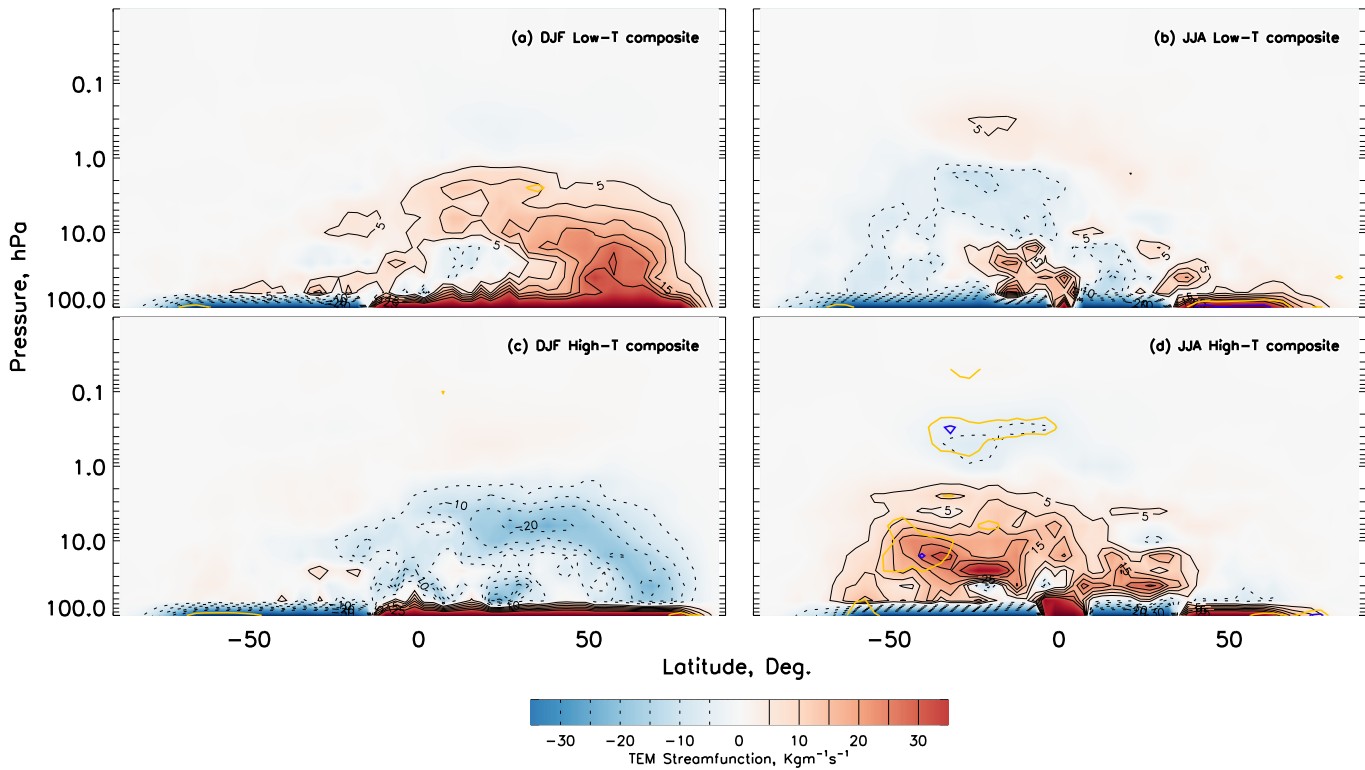

**Figure 5.** The median of the Transformed Eulerian Mean streamfunction (TEMS) anomalies for (a) DJF Low-Temperature events, (b) JJA Low-T events, (c) DJF High-T events, (d) JJA High-T events for the same months as in Fig. 3c–f. Contours lines (solid, positive; dashed negative) and colours are given in the legend. Positive values indicate clockwise-acceleration along the contour lines; negative are anti-clockwise. Data are from the SOCOL model in specified dynamics mode.

We complete the UBDC index by combining November–April NH anomalies with May–October SH anomalies; this combination maximises the relationship with equatorial temperature. We plot the index derived from the CCM SOCOL in specifed dynamics mode using ERA-Interim in Fig. 7. Fig. 8 shows the SH and NH mid-latitude temperature anomalies versus the 4 hPa 20°S–20°N equatorial average (a and b, respectively; grey crosses represent November–April, and black May–October). The
5    SH May–October temperature anomalies are inversely correlated with equatorial temperatures ($r_c$=-0.70) while November–April are not ($r_c$=0.05); the opposite is true for the NH ($r_c$=0.02 and -0.78, respectively). The ozone-temperature events identified in Fig. 1 are highlighted with coloured circles, showing that the equatorial anomalies are related to mid-latitude wave-driving. Fig. 8c shows the UBDC index plotted against all equatorial temperature anomalies at 4 hPa ($r_c$=-0.74). The lower panels (d–f) show the equatorial ozone relationship with respect to mid-latitude temperature and the UBDC index; the
10   absolute correlation coefficient is lower ($r_c$=0.65) than for equatorial temperature in the upper panels, but there is still a strong relationship.

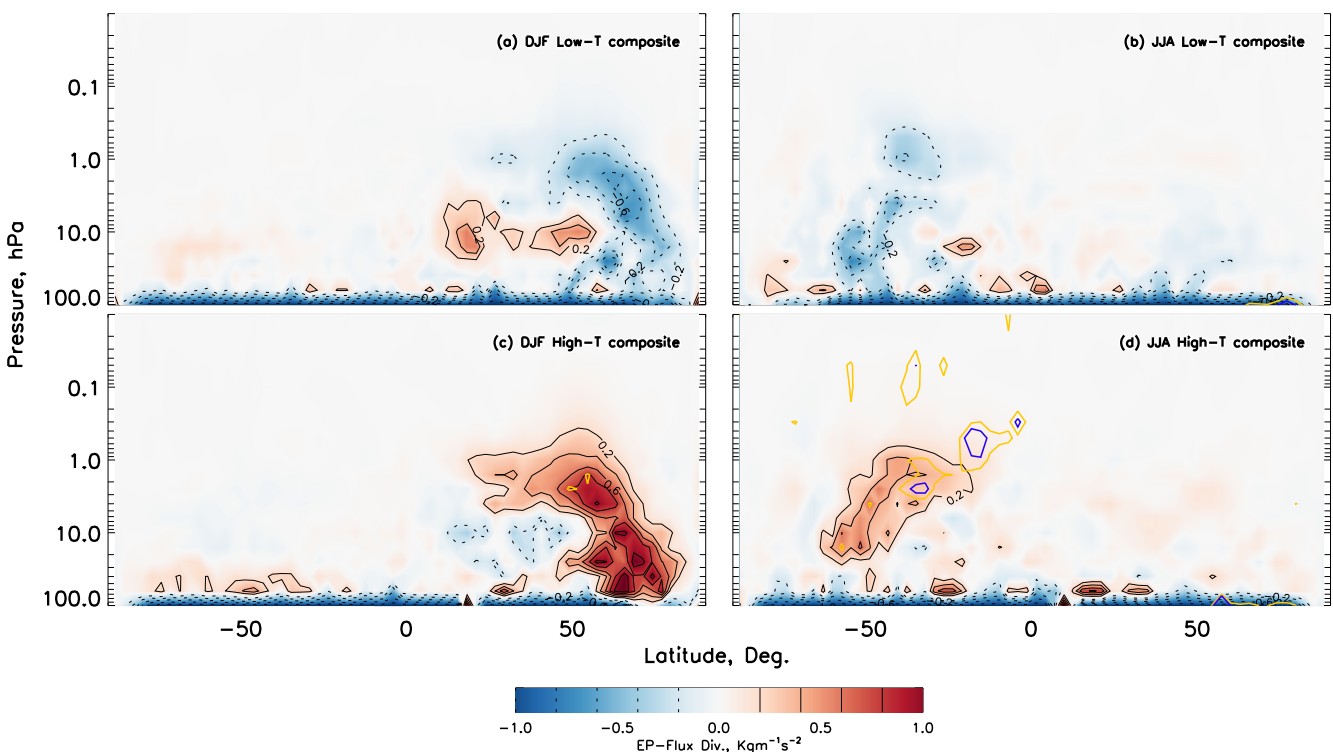

**Figure 6.** As for Fig. 5, but for EP-Flux Divergence. Negative values indicate increased wave-activity; positive, decreased activity.

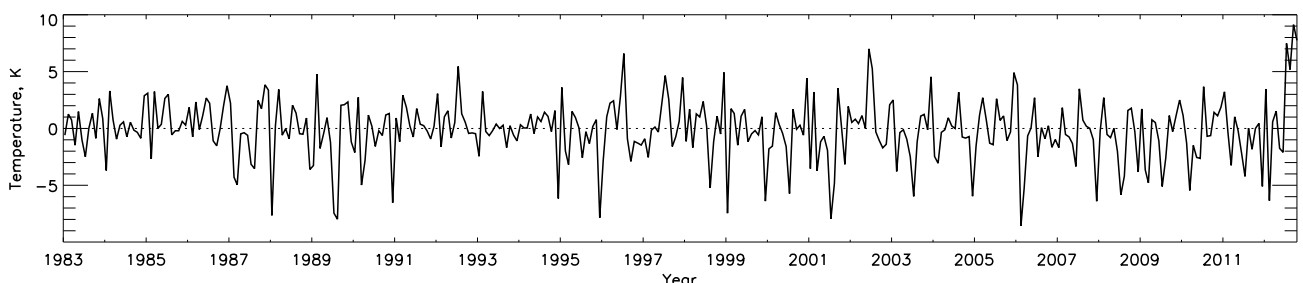

**Figure 7.** The UBDC index from the CCM SOCOL model in specifed dynamics mode using ERA-Interim from 1983 to 2012.

In Fig. 9 we show the amount of variability the UBDC index can account for in nudged-model temperature anomalies everywhere (1983–2012), using the coefficient of determination $r_c^2$, or $R^2$. It ranges from 0 to 1; a value of 1 is synonymous with the index accounting for 100% of the variability. In Fig. 9a the UBDC index can account for >50% of variability between 10 and 1 hPa, and above 0.05 hPa. The variability accounted for at mid-latitudes is less (up to ~30%), even at the index source
5 locations (white circles), because the UBDC index has almost zero agreement half of the time there (see Fig. 8). In Fig. 9b

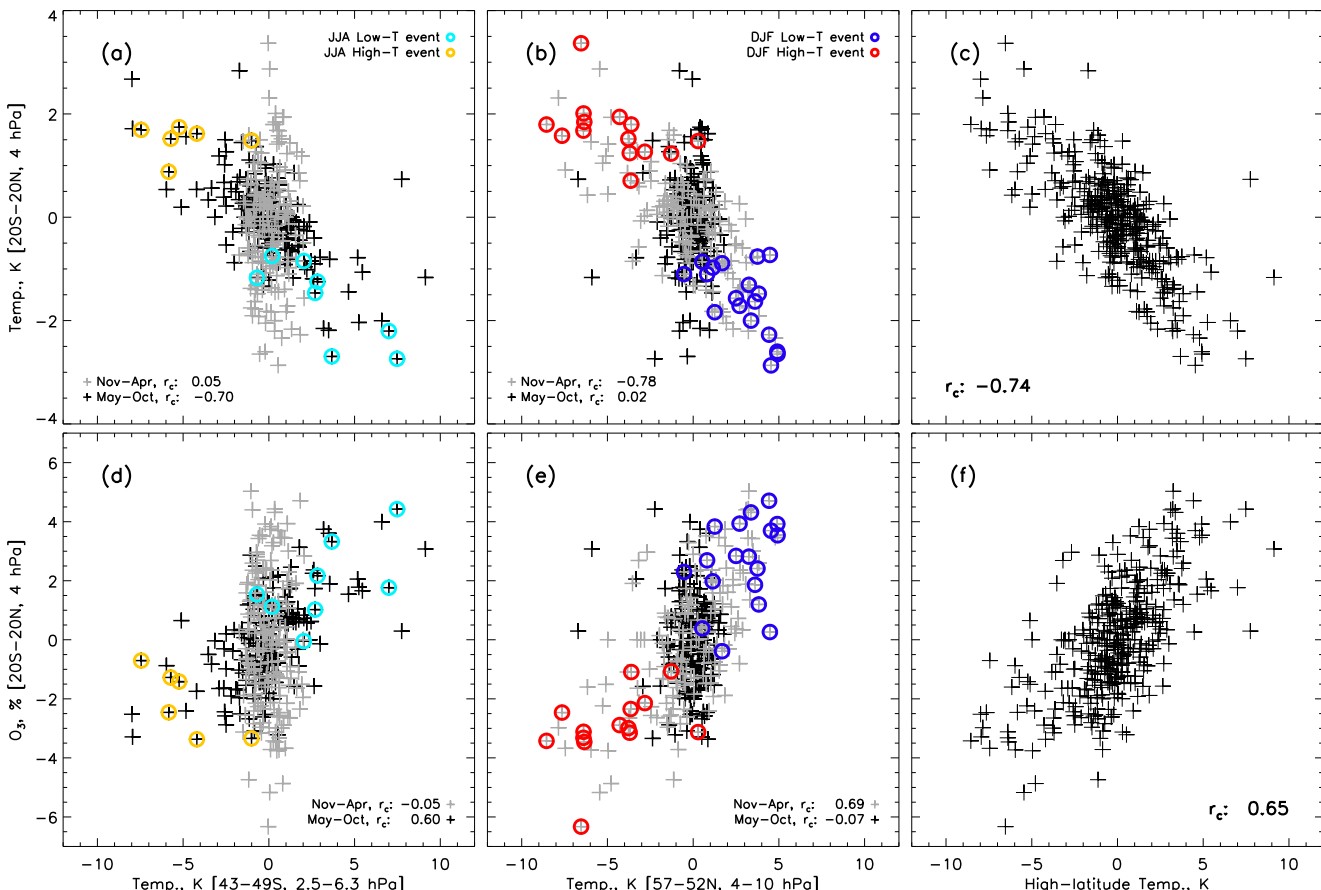

**Figure 8.** (**a–b**) SOCOL equatorial temperature anomalies (4hPa, 20°N–20°S) plotted against (a) temperature means from 2.5–6.3 hPa and 43–49°S and (b) 52–57°N and 4–10 hPa. (**d–e**) As for upper panels, but equatorial ozone anomalies (4 hPa, 20°N–20°S) are instead plotted against high-latitude temperature anomalies. Grey crosses are for November–April months; black crosses for May–October; correlations for both periods are given in each panel. Red and blue circles identify the DJF High-T and Low-T events in Fig. 1 and 2, respectively; orange and light-blue circles similarly identify JJA events. (**c,f**) May–October 43–49°S temperatures and November-April 52–57°N temperatures are combined in the right panel (UBDC index) and plotted against equatorial (c) temperature and (f) ozone.

and c, the UBDC index accounts for much of the DJF/JJA variability: above 20 hPa it can account for over 70% of equatorial variability, more than 60% of polar mesospheric variability (80% in the SH), and much of polar stratosphere variability.

We briefly investigated if there was any indication of an association, through correlation, between the UBDC index and proxies often considered to represent precursors of, or be directly related to, dynamical drivers of lower-stratospheric vari-
5   ability: the North Atlantic Oscillation (NAO) (Hurrell, 1995); the Antarctic Oscillation (AAO) (Marshall, 2003); ENSO; the QBO at both 30 and 50 hPa; and the 100 hPa (eddy) heat flux (Newman et al., 2001) averaged between 60–90°S, 60–90°N, 45–75°N and 45–75°S. We consider DJF and November–April periods for the northern hemisphere, and JJA and May–October

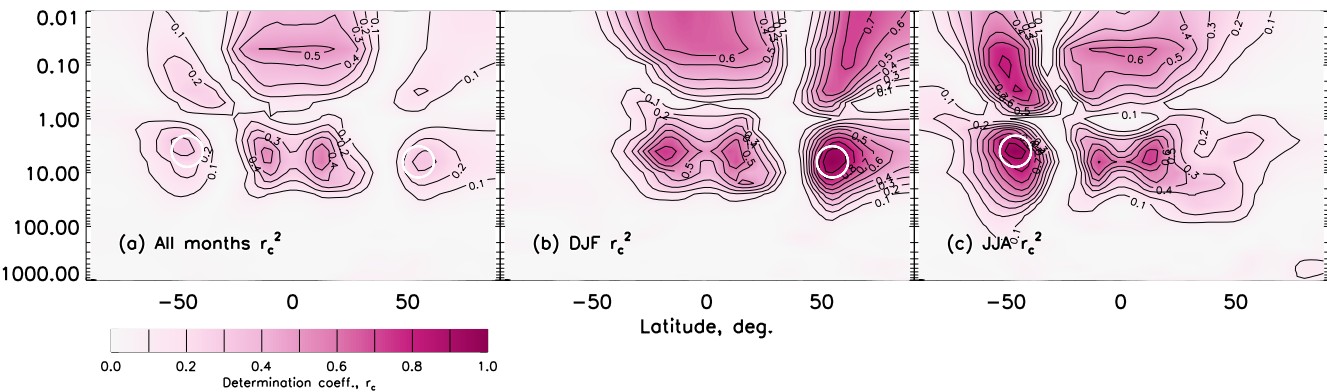

**Figure 9.** Coefficient of determination ($R^2$) maps of the upper-branch Brewer Dobson circulation (UBDC) index with SOCOL model temperature at all latitudes and altitudes for (**a**) all months, (**b**) DJF and (**c**) JJA months. White circles represent the approximate region that the UBDC index is derived from.

for the southern hemisphere. We considered the original timeseries, and detrended and deseasonalised versions following the method outlined in section 3.1. We correlated these 32 'proxy' timeseries with the UBDC index, which we have now shown to have high agreement with short-term variability in the upper-stratosphere and mesosphere. Considering just the $R^2$ values (i.e. coefficient of determination), we found values exceeding 0.15 only for the 100 hPa heat flux in three cases: DJF 60–90°N after

deseasonalising and detrending the data, and 0.18 and 0.19 for DJF 45–75°N, respectively with or without deseasonalising and detrending; we also found that the third case here has a similar value with a 1-month lag. Results for proxies in the southern hemisphere all had values close to zero. While these results suggest there is some possible relationship between the 100 hPa heat flux and dynamical variability in the upper stratosphere, and we concede that this was a simplistic set of tests, the implication is that very little of the variance we see in temperature above 10 hPa is accounted for by these proxies for stratospheric

dynamics at lower altitudes. The source of the upper-stratosphere and mesospheric variance warrants further investigation beyond this publication, as our analysis clearly shows a relationship with changes in temperature in the upper stratosphere and mesosphere related to what appears to be a wave-forcing like response in the EPFD and streamfunctions (i.e. Figs. 5 and 6).

## 5   Improvement in MLR analysis using the UBDC index

The UBDC index leads to a large uncertainty reduction in MLR analysis. To show this, we consider MLR with or without

the index focused on the equatorial region (20°S–20°N). In both cases we use the two QBO indices, SAOD, ENSO, a linear trend (mentioned in section 1), and we use the F30 radio flux as a proxy for solar variability, as this is superior to the F10.7 cm radio flux (Dudok de Wit et al., 2014). **We consider the use of 'AR2' auto-regressive modelling through the procedure of Cochrane and Orcutt (1949) in all cases; see Tiao et al. (1990) for a discussion of AR. The use of second-order autoregression was determined after assessing the regression analysis using a Durbin-Watson test, which showed that AR1**

**was necessary, but not sufficient to account for auto-correlation in the residuals, and that AR2 was sufficient.**. In Fig. 10a,

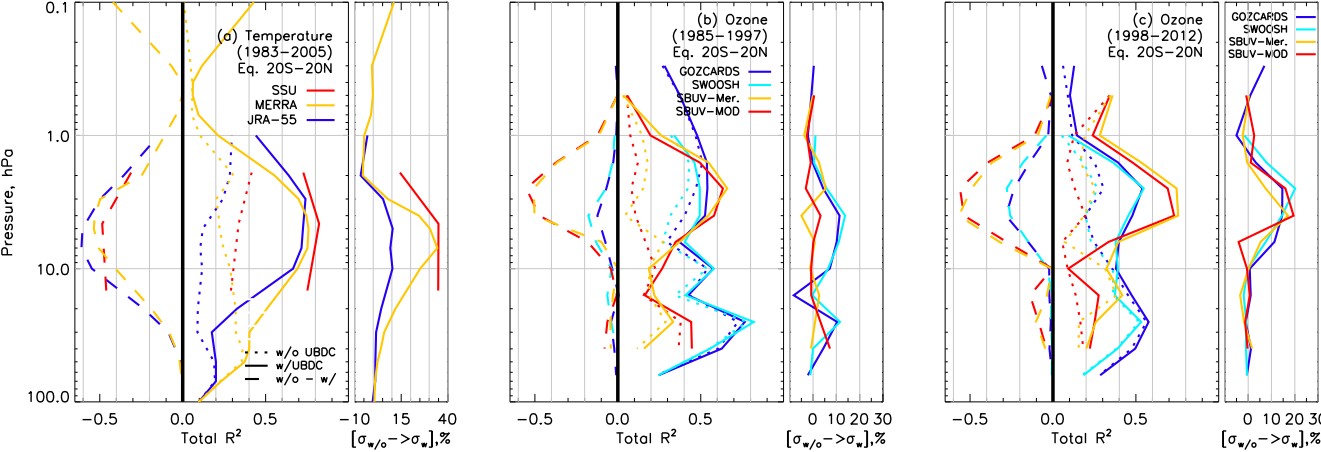

**Figure 10.** Coefficient of determination summed over all regressors ($R_c^2$) and the reduction in the Student's t-test-based error on regressor coefficients (%) for equatorial profiles (positive values) for (**a**) temperature from 1983–2005, and for ozone between (**b**) 1985 and 1997, and (**c**) 1998 and 2012 for various datasets (see legends). For $R^2$, dotted lines represent estimates without the UBDC index, solid lines with, and the difference (without-UBDC minus with-UBDC) is given as negative and dashed lines.

we show the combined ability of the regression model to account for variance, i.e. the total $R^2$ of all regressors, for 1983–2005 in SSU temperature observations (red), JRA-55 (blue) and MERRA (yellow) reanalyses. $R^2$ without the UBDC index (dotted lines; Fig. 10a) shows that only up to ∼45% ($R^2$=0.45) of the stratospheric variability above 10 hPa can be accounted for. However, with the UBDC index (solid lines) $R^2$ is >0.8 (MERRA ∼0.7) and, in all cases, the UBDC index peaks at ∼5 hPa with $R^2$ increasing by 0.45–0.60, or an improvement of up to 60% (see negative values in the left panel of Fig. 10a, i.e. $R^2_{\text{w/o UBDC}}$-$R^2_{\text{w/ UBDC}}$). In the right panel of Fig. 10a, we show the relative change in regressor uncertainty [($\sigma^2_{\text{w/ UBDC}}$ - $\sigma^2_{\text{w/o UBDC}}$) / $\sigma^2_{\text{w/o UBDC}}\times100$], where $\sigma$ is based on the Student's t-test. The uncertainty estimates on the regressors decrease by up to ∼35% (SSU), ∼35% (JRA-55) and ∼10% (MERRA). In addition, the index increases $R^2$ above 0.4 hPa in the mesosphere.

To check whether the UBDC index increases the amount of variance of the total accounted for, in Fig. 11 we calculate the relative importance of each regressor without (blue) and with (red) the UBDC index by decomposing $R^2$ (see Bi (2012) for a comprehensive review of this technique), which depends on the order regressors are considered, unless the regressors are orthogonal, which is usually not the case for the three decades we consider here (see e.g. Chiodo et al. (2014)). We use the robust LGM Measure (Lindeman et al., 1980), which determines relative importance by averaging over all possible, n!, ordering of regressors (720 for 6 regressors, 5040 for 7). In Fig. 11a we show the relative importance of each regressor, and the total, in representing the variance in SSU temperature at 4.6 hPa; curves represent the complete distributions resulting from 10000 bootstrappings of averages over orderings. At 4.6 hPa the UBDC index accounts for ∼61% of temperature variance when considered in addition to the others, partly at the expense of decreasing the relative importance of the other regressors. Together, the UBDC leads to a ∼44% increase in the total variance accounted for, from 38% to 82% (peak values; solid white lines); we see similar results at the other two pressure levels (Fig. 11c). Fig. 11b shows that seasonal MLR analysis is enhanced:

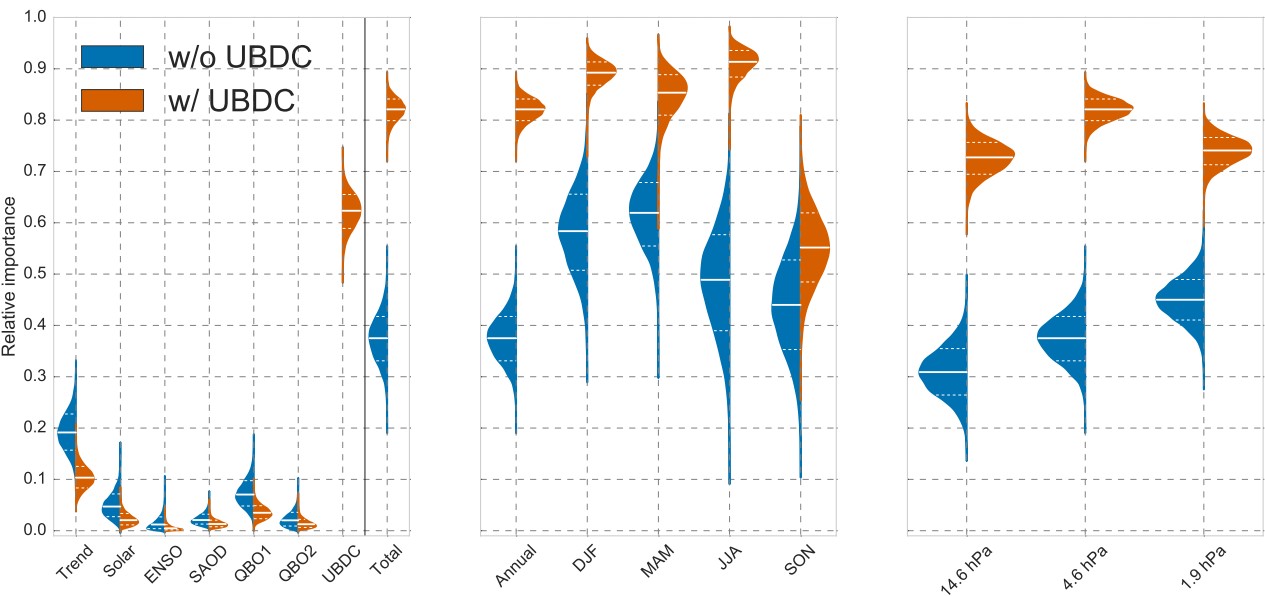

**Figure 11.** The full distributions of $R^2$ from MLR of SSU equatorial temperature (20°S–20°N, 1983–2005) without (blue) and with (red) the UBDC index showing: (**left**) 4.6 hPa $R^2$ values for all the regressors considered in the analysis, and the total; (**middle**) the annual and seasonal total $R^2$ at 4.6 hPa; and (**c**) the annual total $R^2$ for the three SSU pressure levels. Distributions were calculated from 10000 bootstrapped samples for each of the possible (n=6) 720, or (n=7) 5040, order of regressors. Solid white lines are the median values, dotted lines are the 68% confidence intervals.

March-April-May (MAM), JJA and DJF peaks increase by more than double the 68% confidence intervals, i.e. by an additional ∼22%, ∼40% and ∼33%, respectively; September-October-November (SON) months do not improve much (∼10%).

   Similar results are found for ozone. Figs. 10b and c show the ozone composites (see section 2.2) split into 1985–1997 and 1998–2012 time periods, reflecting those often used to investigate ozone trends (e.g. Harris et al. (2015)). There are signifi-
5  cant differences in $R^2$ between the ozone composites from the MLR analysis, which reflects the fact that different equatorial decadal trends are found between the ozone datasets (Tummon et al., 2015; Harris et al., 2015) and in solar signal profiles (Maycock et al., 2016) extracted with MLR, which may be related to the way datasets in the composites have been merged together. While smaller for ozone than temperature, an improvement is found in representing variance ($R^2$ ∼0.55), and errors reduce by up to ∼20% (smaller for the pre-1998 period). While the UBDC index leads to an increase in the variance accounted
10 for in equatorial temperature variability above 40 hPa (Fig. 10a), it only increases above 10 hPa for ozone; this tallies with the strong relationship between temperature and ozone shown in Fig. 2, which also breaks down at 10 hPa. Furthermore, Fig. 12 shows the relative importance of each regressor using violin plots (Waskom et al., 2016), and the total, for all four ozone composites for the 1998–2012 period at 1.6 hPa. The format is the same as for Fig. 11, though we only show relative importance for each regressor, and total, for the case that includes the UBDC index. We see that at this pressure level, most of
15 the variance is given by QBO2 and UBDC indices. The UBDC accounts for between 30 and 55% of the variance in equatorial

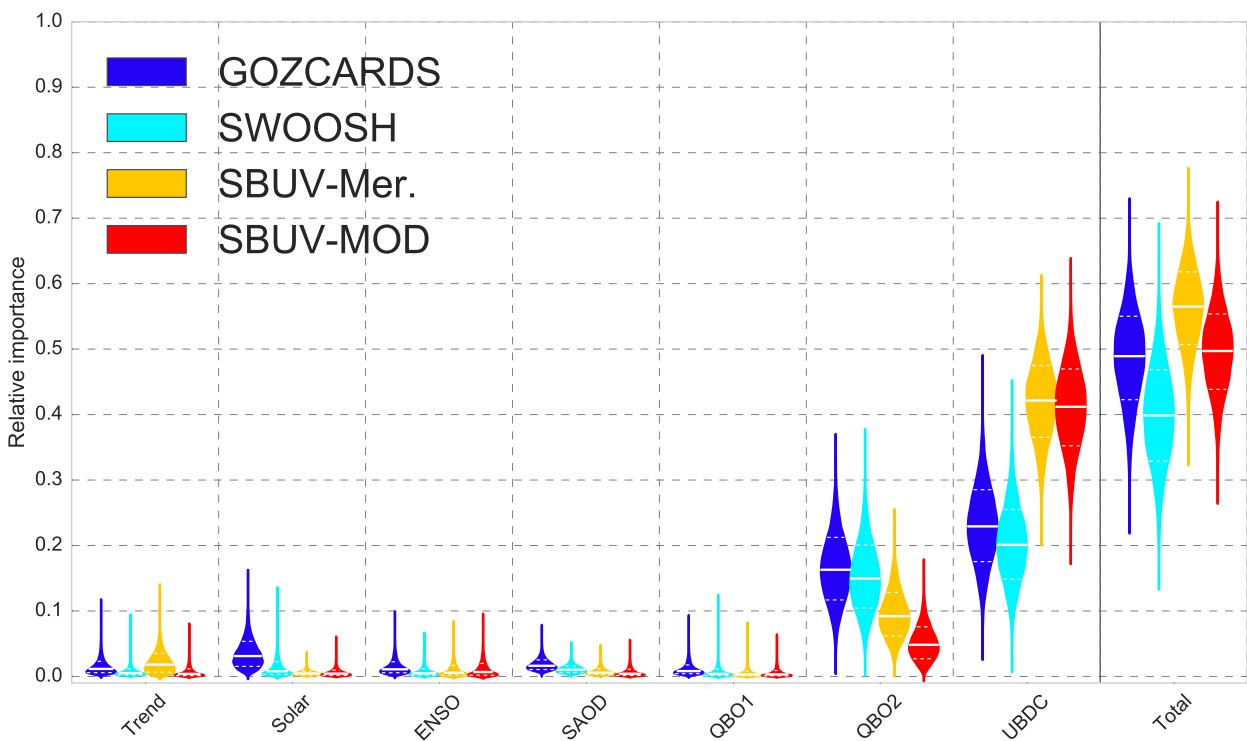

**Figure 12.** Similar in format and method to Fig. 11a, the relative importance of each regressor using the coefficient of determination ($R^2$), and the combined total, are shown from multiple linear regression analysis of four ozone composite datasets at 1.6 hPa for 20°S–20°N and 1998–2012. The most-likely value is given by the central, solid white line, and the dotted lines are the 68% confidence intervals. Distributions were calculated from 10000 bootstrapped samples of each of the possible 5040 regressor orderings.

ozone, depending on altitude and whether the composite is SAGE-II based (GOZCARDS and SWOOSH) or SBUV-based; the results here suggest the data the composites are based upon affects the relative contributions to the total variance (see further discussion below).

We note that the UBDC index influences the relative importance of most of the other regressors by less than the 68% confidence interval (dashed white lines in Fig. 12), and only in the case of the trend does the relative importance get decreased by a larger margin, though within 95%; Fig. 13b shows that the mean values of the SSU MLR results are almost completely unaffected (red dots are with the UBDC index, circles without), and the MERRA and JRA-55 are only marginally affected and therefore the UBDC index does not alias with the estimated trend in temperature. The relative importance of regressors for ozone is affected only slightly in the same way as temperature (not shown), so the larger effect on the SSU relative importance may be data dependent.

In Fig. 13 we show the equatorial decadal trend profiles of the datasets considered in Fig. 10 and the $2\sigma$ uncertainties derived from multiple linear regression with (thick lines) and without (thin lines) the UBDC index, between 25 and 0.2 hPa. A full

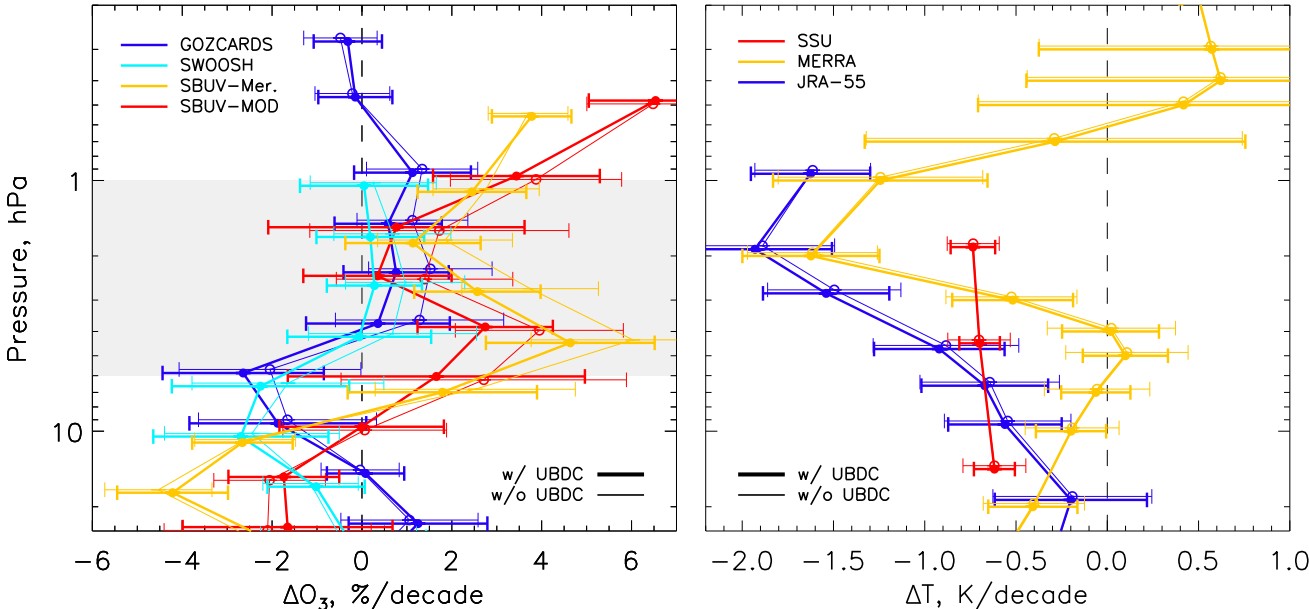

**Figure 13.** Equatorial stratospheric decadal trend profiles for (left) SWOOSH (light-blue), GOZCARDS (blue), SBUV-MER (yellow) and SBUV-MOD (red) ozone between 1998 and 2012, and (right) for SSU (red), MERRA (yellow) and JRA-55 (blue) temperature between 1983 and 2005. Thin lines and circles represent profiles without the UBDC index, thick lines and filled circles are with. The change in error bars are the same reduction in the error bar between using and not using the UBDC index as in Fig. 10. Profiles have been offset slightly from the actual pressure levels for clarity.

discussion of the differences in the ozone profiles is undertaken by Tummon et al. (2015) and Harris et al. (2015), so we do not repeat that here. We simply note that the mean decadal equatorial trends in temperature are affected only slightly by the UBDC index (left panel of Fig. 13). However, we see that the influence of the UBDC index on the mean profile of ozone leads to a decrease in the ozone trend of ∼0.5–1% per decade in all the ozone composites, at the altitudes where the index also performs best at reducing uncertainties (Fig. 13a). This decrease may be a result of the largest anomalies after 1998 being positive (see upper plot in Fig. 1), which might introduce a slight upward bias in the trend analysis; once accounted for with the UBDC index, this bias is removed and the trend is reduced slightly. Nevertheless, this result suggests that ozone trend estimates that do not take the short, anomalous variability into account will overestimate the decadal trends, though it is clear that the biggest uncertainties remain in the underlying datasets themselves (Harris et al., 2015).

## 6   Conclusions

We have shown that detrended and deseasonalised ozone and temperature anomalies in the tropics are strongly influenced by mid-latitude dynamical perturbations that influence temperature throughout the upper stratosphere and mesosphere of the

perturbed hemisphere. The strongest correlations with these anomalies occur at latitudes around 50° in the winter of both hemispheres, which are linked to changes in wave-forcing.

We develop a new upper-branch Brewer Dobson circulation (UBDC) index, which has the power to considerably improve the statistical significance of ozone and temperature trends, and account for much larger fractions of the total variability. **Our results suggest that the index is able to improve the uncertainty of equatorial temperature and ozone trend estimates by up to 35 and 20%, respectively, between 0.5 and 50 hPa and higher in the mesosphere although there is a strong altitude dependence, and up to 60% of the total variance can be accounted for. We also find that this result is data dependent, with the reanalysis products seeing less improvement than the observations**. While we focus on improvements in equatorial temperature and ozone, we suggest it could also be used in the analysis of other stratospheric variables, and also in other regions as well as in the mesosphere. The UBDC index should be employed in future investigations of stratospheric trends in the upper stratosphere and mesosphere. For modelling studies, this index can be extracted from pressure levels and latitudes similar to those put forward here, though the exact peak is likely to be model dependent; for future trends it may be necessary to determine the exact peak again since the regions of wave propagation and breaking may change.

In all cases considered here, the UBDC index both improves our ability to reduce uncertainties and better account for equatorial stratospheric ozone and temperature variability and, by extension, attain better estimates of trends in stratospheric and mesospheric mid-to-high latitude variability.

*Acknowledgements.* We provide all multiple linear analysis results (MLR) on the Mendeley Data portal (Kuchar et al., 2016). We thank A. Y. Karpechko and L. Hood for very helpful suggestions that has led to significant improvements in the quality of the present work. We thank David Thompson for helpful discussion and suggestions. We thank the GOZCARDS, SWOOSH and SBUV teams for their ozone products. We thank the Sounding of the Atmosphere using Broadband Emission Radiometry (SABER/TIMED) science team for their data. We acknowledge the Global Modeling and Assimilation Office (GMAO) and the GES DISC for the dissemination of MERRA, and we acknowledge the Data Integration and Analysis System (DIAS) for the use of JRA-55. We acknowledge SATIRE-S data from http://www2.mps.mpg.de/projects/sun-climate/data.html. W. T. Ball was funded by Swiss National Science Foundation (SNSF) grants 200021_149182 (SILA) and 200020_163206 (SIMA). T. Sukhodolov was funded by SNSF grant 200020_153302. A. Kuchar was funded by Charles University in Prague, No. 1474314; and Czech Science Foundation (GA CR), No. 16-01562J. E. V. Rozanov was partially funded by SNSF grant CRSII2_147659 (FUPSOL-II). F. Tummon was funded by SNSF grant 20F121_138017. The National Center for Atmospheric Research is sponsored by the National Science Foundation.

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
