# Peer review of "An Upper-branch Brewer Dobson Circulation index for attribution of stratospheric variability and improved ozone and temperature trend analysis"

_Atmospheric Chemistry and Physics, 2016_

## Referee Comment (RC1) · A.Yu. Karpechko (Referee) · 1 Aug 2016

Paper's main finding is coherence in the variability of stratospheric temperature and ozone in the tropics and extratropics and in the upper stratosphere and lower mesosphere. The authors attribute this coherence to dynamics, specifically to the stratospheric meridional (Brewer-Dobson) circulation, and propose that an index accounting for dynamical effects could be used in multiple regression analysis as additional regressor. They further build such an index using extratropical upper stratospheric temperatures and demonstrate that the index explains considerable fraction of variability in stratospheric ozone and temperatures. Although the authors present interesting analysis, they still have to show how their analysis is related to previous research and high-

light novel results. The use of regressors accounting for dynamical effects has been discussed in previous WMO Ozone Assessments, discussing their pros and cons. I believe that a more thorough discussion of issues associated with the use of dynamical proxies, as well as relation of the current analysis with previous studies is needed before possible publication in ACP. Please see my specific comments below. Major comments 1. Various dynamical proxies have been used in past to explain stratospheric variability related to dynamics, see examples in Weiss, et al. 2001; Brunner et al. 2006; Mäder et al., 2007; Wohltmann et al., 2005; 2007 and references therein. While a considerable fraction of variability in both ozone and temperatures can indeed be explained by these proxies, this benefit comes at the cost of attributing variability to processes which are themselves dependent on the variables to be explained (wave propagation depends on the mean state of the stratosphere), i.e. one mixes cause and effect. I suggest that these issues should be discussed in the manuscript. Relevant discussion regarding the use of dynamical proxies for attributing ozone variability can be found in Chapter 2 of WMO ozone Assessment 2011 (Sections 2.1.2 and 2.4). References: Brunner, D., J. Staehelin, J.A. Maeder, I. Wohltmann, and G.E. Bodeker, Variability and trends in total and vertically resolved stratospheric ozone based on the CATO ozone data set, Atmos. Chem. Phys., 6 (12), 4985-5008, doi: 10.5194/acp-6-4985-2006, 2006.

Mäder, J.A., J. Staehelin, D. Brunner, W.A. Stahel, I. Wohltmann, and T. Peter, Statistical modeling of total ozone: Selection of appropriate explanatory variables, J. Geophys. Res., 112, D11108, doi: 10.1029/2006JD007694, 2007.

Wohltmann, I., M. Rex, D. Brunner, and J. Mäder (2005), Integrated equivalent latitude as a proxy for dynamical changes in ozone column, Geophys. Res. Lett., 32, L09811, doi:10.1029/2005GL022497.

Wohltmann, I., R. Lehmann, M. Rex, D. Brunner, and J. Mäder, A process-oriented regression model for column ozone, J. Geophys. Res., 112, D12304, doi: 10.1029/2006JD007573, 2007.

Weiss, A. K., J. Staehelin, C. Appenzeller, and N. R. P. Harris (2001), Chemical and dynamical contributions to ozone profile trends of the Payerne (Switzerland) balloon soundings, J. Geophys. Res., 106(D19), 22685–22694, doi:10.1029/2000JD000106 WMO: Scientific Assessment of Ozone Depletion: 2010, Global Ozone Research and Monitoring Project, 52, 516, 2011.

2. There are also problems with using temperature as a proxy representing extratropical wave dynamics. Stratospheric temperature is controlled by a number of processes, such as horizontal and vertical advection, diabatic heating, and not all variability is necessarily directly attributable to extratropical wave forcing. Constructing an index by maximizing correlation, as is done in this study, also maximizes the risk of mixing statistical noise with physical processes. That is why using proxies more directly related to wave activity could be a better choice. While I agree that wave activity proxies such as EP-flux divergence are difficult to calculate, one can try, for example, heat flux evaluated at 100hPa (e.g. Newman et al. 2001), which is quite easy to calculate.

Reference: Newman, P. A., E. R. Nash, and J. E. Rosenfield (2001), What controls the temperature of the Arctic stratosphere during the spring?, J. Geophys. Res., 106(D17), 19999–20010, doi:10.1029/2000JD000061

Other comments: 1. P2L1-5: See Major Comment 1. There are plenty of studies using different set of proxies, not only the six proxies listed here. 2. P2L118: I believe there are older references which show influence of dynamics on stratospheric ozone, e.g. Fusco and Salby 1999 and references therein.

Reference: Fusco, A. C. and Salby, M. L.: Interannual variations of total ozone and their relationship to variations of planetary wave activity, J. Clim., 12, 1619–1629, 1999.

3. P2L22-23: Please note that acceleration of BD circulation leads not only to increase of ozone in the extratropics but also to a decrease in the tropics, thus it is more correct to say that ozone is redistributed, not just increased.

4. P3L27-28: I think smoothing removes short-term variability, not long-term. Please rewrite.

5. P8L4-6: Please see Major Comment 2. I think some caution is needed when using stratospheric temperature as proxy for dynamics.

6. P9L6: 'Verses' -> 'versus'

7. Figure 10: The difference in Fig. 10b between regression results from GOZCARDS and SWOOSH from the one hand and SBUV from the other hand are interesting. It appears like dynamical variability in GOZCARDS and SWOOSH is represented by the other proxies because, after addition of the dynamical proxy, the explained variability changes only little in these data sets, and the total explained variability is quite similar in all four data sets. Do you think it is purely statistical effect or it may be related to the way these data sets are compiled? (Sorry I am not familiar with these data sets.)

8. Figure 11: I am puzzled by why the annual $R^2$ for the w/ MLDS regression in the middle panel is larger than any seasonal one. The result from the w/o regression, where the annual $R^2$ looks like the mean of seasonal results, looks more logical, is it not?

9. Captions to Figure 11: What is distribution peak? Is it the mode?

---

## Referee Comment (RC2) · L. Hood (Referee) · 2 Aug 2016

Overall, this is a useful effort to improve statistical estimation of stratospheric ozone and temperature trends and interannual variability by accounting for a source of short-term (month-to-month) dynamical variability in tropical stratospheric data sets. The presentation is excellent and the figures are state-of-the art. However, the value of the adopted technique for trend estimation and its ability to "explain" a larger fraction of the variance in the observations is somewhat overstated, in my opinion. Some important revisions are needed prior to publication.

Main comments:

[Figure]

(1) A major claim of the paper is that inclusion of the mid-latitude stratosphere dynamical (MLSD) index can reduce the uncertainty "on all multiple linear regression coefficients ... up to 45% and 25% in temperature and ozone, respectively." First of all, the accuracy of these reduction estimates is questionable because, as mentioned on p. 11, line 12, "we do not consider use of any autoregressive modeling." In other words, serial correlation (autocorrelation) of the residuals of the MLR analysis is not accounted for. It is possible that serial correlation of the monthly residuals is increased when the MLSD index is used because the month-to-month variability is reduced. Have the authors tested whether this is the case? Accounting for any increased serial correlation would increase the uncertainty estimates. For example, application of a "pre-whitening" technique (e.g., Tiao et al. [1990]; Garny et al. [2007]) would ensure that the residuals are approximately white noise thereby yielding more reliable uncertainty estimates. Please re-do the analysis in this manner to provide such a test and yield more accurate (larger) uncertainty estimates. Second, even without accounting for serial correlation, the difference in the ozone and temperature trend results with and without the MLSD term shown in Figure 13 is not very impressive. For the sake of clarity, consider only the yellow curves in the figure. The error bars for the with MLSD (thick curves) and without MLSD (thin curves) cases overlap. These are presumably $2\sigma$ error bars, right? If not, then the overlap is even larger. The error bars are roughly the same size at most levels. At 2.5 hPa, the ozone error bar appears to be about 25% smaller for the with MLSD case, which is consistent with the authors' statement. But it is not a very significant difference considering the sizes of the error bars and the large variation in the trend estimates from one pressure level to the next. For most of the other levels, the difference in size of the error bars is hard to discern.

(2) The other major claim of the paper is that use of the MLSD index in a regression analysis can "explain much larger fractions of the total variability." I am not sure that the word "explain" is appropriate. The dynamically induced variability is being accounted for in the MLR analysis but it is not really being explained. For example, the

see-saw temperature and ozone variations between the tropics and extratropics are in many cases associated with minor and major polar stratospheric warmings in the winter hemispheres. The latter are modulated by a number of external forcings including the QBO and the solar cycle. A true explanation of the variability would therefore need to account for the external forcings that are controlling the rate of wave absorption events, which in turn produce the ozone and temperature fluctuations. I also disagree with the terminology "total coefficient of determination", which is used in place of explained variance ($R^2$) in the text. The words "determination", "explained", and "attribution" are all misleading if the sources of the dynamical fluctuations are not identified. Please revise the introduction and conclusions section to make this clear.

Minor comments:

(3) I agree with the other referee that the history of the ozone and temperature variations that are discussed in the paper and their application to trend analyses is not adequately summarized in the paper. The first report of the existence of such global stratospheric temperature oscillations with a change in phase between low and middle to high latitudes was by Fritz and Soules [1970]. Some stratospheric dynamicists still refer to these oscillations as the "Fritz-Soules effect". See also, e.g., Andrews et al. [1987] for general discussions of their dynamical origin. Another observational study by Chandra [1986] could also be referenced.

(4) In Figure 1 (and maybe other figures), the definitions of the diamonds in the upper right corner seem to be incorrect and are opposite to those given in the caption.

(5) P. 7, line 7. adiabatically

References:

Andrews, D. G., J. R. Holton, and C. B. Leovy, *Middle Atmosphere Dynamics* Academic

[Figure]

Press, 489 pp., 1987.

Chandra, S., The solar and dynamically induced oscillations in the stratosphere, *J. Geophys. Res., 91*, 2719-2734, 1986.

Fritz, S., and S. D. Soules, Large-scale temperature changes in the stratosphere observed from Nimbus-3, *J. Atmos. Sci., 27*, 1091, 1970.

Garny, H., G. E. Bodeker, and M. Dameris, Trends and variability in stratospheric mixing: 1979-2005, *Atmos. Chem. Phys., 7*, 5611-5624.

Tiao, G., et al., Effects of autocorrelation and temporal sampling schemes on estimates of trend and spatial correlation, *J. Geophys. Res., 95*, 20507-20517, 1990.

---

## Author Comment (AC1) · 19 Oct 2016

**Response to referee interactive comments on "A mid-latitude stratosphere dynamical index for attribution of stratospheric variability and improved ozone and temperature trend analysis" by William T. Ball et al**

**General comments relevant to both referees:**

We thank both referees for very helpful guidance and suggestions in the background, justification and statistical analysis performed. This has led to significant improvements in the quality of the manuscript (which are already apparent in our current revisions and in our response below). Major changes in the revised manuscript have been highlighted in bold text.

We begin by highlighting major updates that both reviewers should be aware of:

i) Figures 10, 11, 12 and 13 have been updated to reflect the use of AR1 instead of AR0. Figure 13 in particular has been completely changed, and numbers added, to clarify the error improvement and a change in the mean values. Figure 12 has been updated to reflect similar style and information as provided in Figure 11.

ii) The point made in the manuscript about no aliasing between regressors being shown by the relative importance plots has been modified. Due to the use of AR1, for temperature, there is a redistribution of the relative importance from the original regressors without the new index to the new one, in addition to the increase in total variance accounted for. However, the fact it does not change the mean value of the regression coefficients in the trend still supports the claim that it does not alias the derived signal. Modifications to the text in the manuscript have been made to reflect this change, and discussion on this point has been added.

iii) A significant amount of background have been added to the introduction.

iv) We have renamed the MLSD index to the Upper-branch Brewer Dobson Circulation (UBDC) index, to reflect a more direct interpretation of what it represents, and for which it should be more easily understood; this changes the title of the manuscript.

**A. Karpechko (Referee)**

Paper's main finding is coherence in the variability of stratospheric temperature and ozone in the tropics and extratropics and in the upper stratosphere and lower mesosphere. The authors attribute this coherence to dynamics, specifically to the stratospheric meridional (Brewer-Dobson) circulation, and propose that an index accounting for dynamical effects could be used in multiple regression analysis as additional regressor. They further build such an index using extratropical upper stratospheric temperatures and demonstrate that the index explains considerable fraction of variability in stratospheric ozone and temperatures. Although the authors present interesting analysis, they still have to show how their analysis is related to previous research and high light novel results. The use of regressors accounting for dynamical effects has been discussed in previous WMO Ozone Assessments, discussing their pros and cons. I believe that a more thorough discussion of issues associated with the use of dynamical proxies, as well as relation of the current analysis with previous studies is needed before possible publication in ACP. Please see my specific comments below.

Major comments 1. Various dynamical proxies have been used in past to explain stratospheric variability related to dynamics, see examples in Weiss, et al. 2001; Brunner et al. 2006; Mäder et al.,

2007; Wohltmann et al., 2005; 2007 and references therein. While a considerable fraction of variability in both ozone and temperatures can indeed be explained by these proxies, this benefit comes at the cost of attributing variability to processes which are themselves dependent on the variables to be explained (wave propagation depends on the mean state of the stratosphere), i.e. one mixes cause and effect. I suggest that these issues should be discussed in the manuscript. Relevant discussion regarding the use of dynamical proxies for attributing ozone variability can be found in Chapter 2 of WMO ozone Assessment 2011 (Sections 2.1.2 and 2.4).

We agree, and we appreciate the useful set of references that has led to an expansion of the discussion in the manuscript. Further, the new background material simply further highlights the need for such a dynamical proxy, especially in the equatorial region, since previous studies have focused on BDC proxies that operate on inter-annual timescales and longer, and while often being briefly mentioned, the monthly and shorter timescales are usually ignored (except, e.g. Chandra et al., 1986 as mentioned by the second reviewer). Previous studies have also focused on total ozone column, mid-to-high latitudes, and the mid-to-lower stratosphere. This leads to the clear conclusion that, while not a new concept, the development of a proxy accounting for noise-like dynamical events in the upper stratosphere and mesosphere is necessary, and that its application and focus on the equatorial region is new.

*References:*

1. *Brunner, D., J. Staehelin, J.A. Maeder, I. Wohltmann, and G.E. Bodeker, Variability and trends in total and vertically resolved stratospheric ozone based on the CATO ozone data set, Atmos. Chem. Phys., 6 (12), 4985-5008, doi: 10.5194/acp-6-4985-2006, 2006.*
2. *Mäder, J.A., J. Staehelin, D. Brunner, W.A. Stahel, I. Wohltmann, and T. Peter, Statistical modeling of total ozone: Selection of appropriate explanatory variables, J. Geophys. Res., 112, D11108, doi: 10.1029/2006JD007694, 2007.*
3. *Wohltmann, I., M. Rex, D. Brunner, and J. Mäder (2005), Integrated equivalent latitude as a proxy for dynamical changes in ozone column, Geophys. Res. Lett., 32, L09811, doi:10.1029/2005GL022497.*
4. *Wohltmann, I., R. Lehmann, M. Rex, D. Brunner, and J. Mäder, A processoriented regression model for column ozone, J. Geophys. Res., 112, D12304, doi: 10.1029/2006JD007573, 2007.*
5. *Weiss, A. K., J. Staehelin, C. Appenzeller, and N. R. P. Harris (2001), Chemical and dynamical contributions to ozone profile trends of the Payerne (Switzerland) balloon soundings, J. Geophys. Res., 106(D19), 22685–22694, doi:10.1029/2000JD000106*
6. *WMO: Scientific Assessment of Ozone Depletion: 2010, Global Ozone Research and Monitoring Project, 52, 516, 2011.*

2. There are also problems with using temperature as a proxy representing extratropical wave dynamics. Stratospheric temperature is controlled by a number of processes, such as horizontal and vertical advection, diabatic heating, and not all variability is necessarily directly attributable to extratropical wave forcing. Constructing an index by maximizing correlation, as is done in this study, also maximizes the risk of mixing statistical noise with physical processes. That is why using proxies more directly related to wave activity could be a better choice. While I agree that wave activity proxies such as EP-flux divergence are difficult to calculate, one can try, for example, heat flux evaluated at 100hPa (e.g. Newman et al. 2001), which is quite easy to calculate.

Again, we agree with this assessment and indeed we will make it clear that we are mixing in physical processes from different specific sources (see previous response) in the introduction. Further, it would make sense that if wave-driving from the troposphere at mid-latitudes is one of the main drivers of variance in the equatorial upper stratosphere, then the use of EP-flux divergence (EPFD), or the heat flux at 100 hPa would represent a more physical proxy. We note, however, that while

Newman et al. (2001) were successful in representing short-term dynamical fluctuations in the stratosphere with EPFD, they did not investigate effects above 10 hPa. During the analysis of our original manuscript, we investigated the relationship between 100 hPa heat flux and equatorial ozone and temperature in the upper stratosphere, but were unable to find any clear agreement, even if we considered a lag to account for the time waves take to propagate into the stratosphere and force a response. We revisited this following the reviewer's suggestion (above) and considered the correlation between indices of the NAO, AAO, ENSO, QBO, and 100 hPa heat flux (v'T') averaged between 60-90 S, 60-90 N, 45-75 N and 45-75 S. We further divided months out to consider Dec-Feb and Nov-Apr for the northern hemisphere, and Jun-Aug and May-Oct for the southern hemisphere, and either the original timeseries or detrended and deseasonalised. We compared all of these cases with the MLSD index, which has high agreement locally to the respective hemisphere and with the equatorial upper stratosphere and mesosphere. Considering just the $R^2$ values (i.e. correlation coefficient squared), we found agreement exceeding 0.15 only in three cases: for DJF 60-90 N after deseasonalising and detrending the data, and 0.18 and 0.19 for DJF 45-75 N with and without deseasonalising and detrending (the third value shown is also similar for a 1 month lag); there was nothing clear in the southern hemisphere, with values all close to zero. This indeed suggests that there is possibly some relationship between heat flux at 100 hPa, and we concede this was a simplistic set of checks. However, the three results showing some coherence with the MLSD index account for very little of the variance we see in temperature above 10 hPa. The source of the variance warrants further investigation beyond this manuscript, as our analysis clearly shows a relationship with changes in temperature in the upper stratosphere and mesosphere related to what appears to be a wave-forcing like response in the EPFD and stream functions (i.e. Figs 5 and 6).

*Reference: Newman, P. A., E. R. Nash, and J. E. Rosenfield (2001), What controls the temperature of the Arctic stratosphere during the spring?, J. Geophys. Res., 106(D17), 19999–20010, doi:10.1029/2000JD000061*

**Other comments:**

1. P2L1-5: See Major Comment 1. There are plenty of studies using different set of proxies, not only the six proxies listed here.

We have expanded this section to a separate discussion of equatorial and higher latitude MLR analysis, which, e.g. includes other proxies such as the AO and NAO, but also see our response to Major Comment 1.

2. P2L118: I believe there are older references which show influence of dynamics on stratospheric ozone, e.g. Fusco and Salby 1999 and references therein.

We have included this reference in addition to others, and those already mentioned above.

*Reference: Fusco, A. C. and Salby, M. L.: Interannual variations of total ozone and their relationship to variations of planetary wave activity, J. Clim., 12, 1619–1629, 1999.*

3. P2L22-23: Please note that acceleration of BD circulation leads not only to increase of ozone in the extratropics but also to a decrease in the tropics, thus it is more correct to say that ozone is redistributed, not just increased.

We suggest this can be made clear with the addition of the following bold text:
*"The increase in ozone at mid-latitudes comes partly from ODSs reductions, but also because the BDC is expected to accelerate (Garcia and Randel, 2008; Butchart, 2014), which will reduce the time for*

*ozone depletion to occur and lead to faster transport of ozone from the equatorial region to higher latitudes. This in turn leads to a reduction of ozone over the equator and a prevention of a full recovery over the tropics.* **Thus, the recovery of ozone at mid-to-high latitudes can be understood as being partly due to less ozone destruction by lower ODS concentrations, and partly due to a faster redistribution of ozone-rich air from the tropics."**

4. P3L27-28: I think smoothing removes short-term variability, not long-term. Please rewrite.

The sentence was incorrectly formulated, and should make clear we remove the smoothed time series from the original; thus the following formulation should be clearer.

*"…we remove all long-term variability by subtracting a timeseries that has been smoothed, with a 13-month running mean, and then deseasonalised, with monthly values, at each latitude and pressure."*

5. P8L4-6: Please see Major Comment 2. I think some caution is needed when using stratospheric temperature as proxy for dynamics.

See response to major comments 1 and 2 and revisions of the text.

6. P9L6: 'Verses' -> 'versus'

Done

7. Figure 10: The difference in Fig. 10b between regression results from GOZCARDS and SWOOSH from the one hand and SBUV from the other hand are interesting. It appears like dynamical variability in GOZCARDS and SWOOSH is represented by the other proxies because, after addition of the dynamical proxy, the explained variability changes only little in these data sets, and the total explained variability is quite similar in all four data sets. Do you think it is purely statistical effect or it may be related to the way these data sets are compiled? (Sorry I am not familiar with these data sets.)

This is an entirely valid, and interesting, question. As you suggest, we believe (and have evidence to support) that the answer resides in the way the datasets are compiled. Indeed, looking at individual time series, it becomes clear that the earlier periods in GOZCARDS and SWOOSH at these altitudes contain high variance fluctuations that look more related to the datasets used themselves than real variability; GOZCARDS and SWOOSH use a similar source of data (SAGE II) for this period. Given the variance is on the order of that in the MLSD index, it is likely (and this is now a postulation) that the reduced improvement during this period is due to these high variance artefacts. We are tackling this problem and are due to submit an article relating to this soon.

8. Figure 11: I am puzzled by why the annual R2 for the w/ MLDS regression in the middle panel is larger than any seasonal one. The result from the w/o regression, where the annual R2 looks like the mean of seasonal results, looks more logical, is it not?

As you correctly identified, there was a mistake in how anomalies were dealt with in the regression routine. This has been corrected, and indeed the results now appear more logical.

9. Captions to Figure 11: What is distribution peak? Is it the mode?

The peak is, more precisely, the median and we have added this to the description.

---

## Author Comment (AC2) · 19 Oct 2016

**Response to referee interactive comments on "A mid-latitude stratosphere dynamical index for attribution of stratospheric variability and improved ozone and temperature trend analysis" by William T. Ball et al**

**General comments relevant to both referees:**

We thank both referees for very helpful guidance and suggestions in the background, justification and statistical analysis performed. This has led to significant improvements in the quality of the manuscript (which are already apparent in our current revisions and in our response below). Major changes in the revised manuscript have been highlighted in bold text.

We begin by highlighting major updates that both reviewers should be aware of:

i) Figures 10, 11, 12 and 13 have been updated to reflect the use of AR1 instead of AR0. Figure 13 in particular has been completely changed, and numbers added, to clarify the error improvement and a change in the mean values. Figure 12 has been updated to reflect similar style and information as provided in Figure 11.

ii) The point made in the manuscript about no aliasing between regressors being shown by the relative importance plots has been modified. Due to the use of AR1, for temperature, there is a redistribution of the relative importance from the original regressors without the new index to the new one, in addition to the increase in total variance accounted for. However, the fact it does not change the mean value of the regression coefficients in the trend still supports the claim that it does not alias the derived signal. Modifications to the text in the manuscript have been made to reflect this change, and discussion on this point has been added.

iii) A significant amount of background have been added to the introduction.

iv) We have renamed the MLSD index to the Upper-branch Brewer Dobson Circulation (UBDC) index, to reflect a more direct interpretation of what it represents, and for which it should be more easily understood; this changes the title of the manuscript.

*L. Hood (Referee)*

Overall, this is a useful effort to improve statistical estimation of stratospheric ozone and temperature trends and interannual variability by accounting for a source of shortterm (month-to-month) dynamical variability in tropical stratospheric data sets. The presentation is excellent and the figures are state-of-the art. However, the value of the adopted technique for trend estimation and its ability to "explain" a larger fraction of the variance in the observations is somewhat overstated, in my opinion. Some important revisions are needed prior to publication.

**Main comments:**

(1) A major claim of the paper is that inclusion of the mid-latitude stratosphere dynamical (MLSD) index can reduce the uncertainty "on all multiple linear regression coefficients ... up to 45% and 25% in temperature and ozone, respectively." First of all, the accuracy of these reduction estimates is questionable because, as mentioned on p. 11, line 12, "we do not consider use of any autoregressive modeling." In other words, serial correlation (autocorrelation) of the residuals of the MLR analysis is not accounted for. It is possible that serial correlation of the monthly residuals is increased when the MLSD index is used because the month-to-month variability is reduced. Have the authors tested

whether this is the case? Accounting for any increased serial correlation would increase the uncertainty estimates. For example, application of a "pre-whitening" technique (e.g., Tiao et al. [1990]; Garny et al. [2007]) would ensure that the residuals are approximately white noise thereby yielding more reliable uncertainty estimates. Please re-do the analysis in this manner to provide such a test and yield more accurate (larger) uncertainty estimates.

These are indeed important points. Serial correlation is important, and you are correct that their consideration does indeed puff-up error bars. However, it does not change the main result and the usefulness of the index. To make this point clear we have produced a new plot, which we include in the paper, to emphasize that auto-regression will have an effect and should be considered. Further, we will replace Fig 13 with this one, since the point of Fig 13 is to show clearly how the reduction (and any effect on mean value) works in practice – this also addresses the second main point below. The plot is shown below: AR0 (blue) and AR1 (yellow) are shown for cases with (thick lines) and without (thin) the MLSD index for SWOOSH (ozone, left) and SSU (Temp., right). We see the percentage change (in respective colours) at heights where we see the largest changes. In both cases, we still have a maximum improvement of up to 30% in the errors. Not shown here, but will be in the final manuscript, is that the index now increases $R^2$ from a maximum increase of 40% (fig 10), to nearly 60% in temperature, and between 30 and 55% for ozone (depending on the dataset used). In the 1998-2012 periods, ozone error improvements are essentially unaffected at a 25% reduction in uncertainty, but the earlier 1983-1997 period is affected, on average reducing uncertainties by around 10% to a maximum of around 15%; some regions show a small increase in error, but this likely reflects the fact the datasets show different variability on all timescales (see response to question from the other reviewer on this point). We will update the manuscript to reflect this. [Additional note: we also considered AR2, but AR1 was sufficient to account for partial correlation at 1-month]

[Figure]

Second, even without accounting for serial correlation, the difference in the ozone and temperature trend results with and without the MLSD term shown in Figure 13 is not very impressive. For the sake of clarity, consider only the yellow curves in the figure. The error bars for with (thick curves) and without (thin curves) the MLSD cases overlap. These are presumably 2σ error bars, right? If not, then the overlap is even larger. The error bars are roughly the same size at most levels. At 2.5 hPa, the ozone error bar appears to be about 25% smaller for the with MLSD case, which is consistent with the authors' statement. But it is not a very significant difference considering the sizes of the error

bars and the large variation in the trend estimates from one pressure level to the next. For most of the other levels, the difference in size of the error bars is hard to discern.

We agree with these comments (the error bars are 2-sigma). In fact, we tried to make this clear with the grey shading in the old version of Fig 13a to highlight the altitudes where we see the largest improvement. In hindsight, the plot has such a large absolute range of profiles, that seeing this improvement is difficult. Figure 10 already shows similar results, that is the reduction in uncertainty as a function of altitude (right panels of each sub-plot) – the idea of Figure 13 was to show how it appeared in practice. By accounting for an attributable source of variability (or at least being able to show that it is not simply noise, but a clear dynamical factor) we make a step closer to better understanding those variables we are trying to determine (e.g. trend and solar cycle) – see point below. The new figure (above) reduces the absolute range and focuses in on one of the datasets. We consider this a more useful plot, and discuss and refer to other articles that do show the profiles.

(2) The other major claim of the paper is that use of the MLSD index in a regression analysis can "explain much larger fractions of the total variability." I am not sure that the word "explain" is appropriate. The dynamically induced variability is being accounted for in the MLR analysis but it is not really being explained. For example, the see-saw temperature and ozone variations between the tropics and extratropics are in many cases associated with minor and major polar stratospheric warmings in the winter hemispheres. The latter are modulated by a number of external forcings including the QBO and the solar cycle. A true explanation of the variability would therefore need to account for the external forcings that are controlling the rate of wave absorption events, which in turn produce the ozone and temperature fluctuations. I also disagree with the terminology "total coefficient of determination", which is used in place of explained variance (R2) in the text. The words "determination", "explained", and "attribution" are all misleading if the sources of the dynamical fluctuations are not identified. Please revise the introduction and conclusions section to make this clear.

We are happy to clear up terminology. As the other reviewer also pointed out, the use of temperature mixes potential sources of the variance that correlates with temperature, but is actually the underlying driver, and we have added additional text to the introduction to account for this. The point we are trying to make is that we can 'account' for variability that is physical, and not simply noise that, unconsidered, would lead to higher uncertainty in quantities we wish to determine. It is true, the index itself doesn't necessarily represent the underlying driver of the changes in the meridional flow, but it does act as a proxy and is related to a real variance in the system (which we relate through the EPFD to wave driving, as shown in the manuscript). We disagree about the use of the coefficient of determination, $R^2$, and would argue it is a useful quantity with which to test how much better our regression model, with the index, improves the amount of variability we can account for. By applying the bootstrapping (examples in Figs 11 and 12), we can also account for further statistical uncertainties to ensure that the improvement from the additional dynamical index is robust.

**Minor comments:**

(3) I agree with the other referee that the history of the ozone and temperature variations that are discussed in the paper and their application to trend analyses is not adequately summarized in the paper. The first report of the existence of such global stratospheric temperature oscillations with a change in phase between low and middle to high latitudes was by Fritz and Soules [1970]. Some stratospheric dynamicists still refer to these oscillations as the "Fritz-Soules effect". See also, e.g., Andrews et al. [1987] for general discussions of their dynamical origin. Another observational study by Chandra [1986] could also be referenced.

We have added additional discussion and references as suggested by both referees (see response above to the first referee on this point). The reference by Chandra [1986] was particularly enlightening; our findings also confirm, and expand upon, the results from that study.

(4) In Figure 1 (and maybe other figures), the definitions of the diamonds in the upper right corner seem to be incorrect and are opposite to those given in the caption.

You are correct: the legend in the figure was wrong; this has been fixed; we also checked the other figures, which did not have this problem.

(5) P. 7, line 7. Adiabatically

We have corrected this.

*References:*

1. *Andrews, D. G., J. R. Holton, and C. B. Leovy, Middle Atmosphere Dynamics Academic Press, 489 pp., 1987.*
2. *Chandra, S., The solar and dynamically induced oscillations in the stratosphere, J. Geophys. Res., 91, 2719-2734, 1986.*
3. *Fritz, S., and S. D. Soules, Large-scale temperature changes in the stratosphere observed from Nimbus-3, J. Atmos. Sci., 27, 1091, 1970.*
4. *Garny, H., G. E. Bodeker, and M. Dameris, Trends and variability in stratospheric mixing: 1979-2005, Atmos. Chem. Phys., 7, 5611-5624.*
5. *Tiao, G., et al., Effects of autocorrelation and temporal sampling schemes on estimates of trend and spatial correlation, J. Geophys. Res., 95, 20507-20517, 1990*

---

## Author Response (AR2)

**Response to non-interactive referee comments on "A mid-latitude stratosphere dynamical index for attribution of stratospheric variability and improved ozone and temperature trend analysis" by William T. Ball et al**

**General comments relevant to both referees:**

We thank both reviewers for their continuing input. Please see our comments (blue) below in response to the reviewers (black). Any major changes to the text (see below) have been put in bold font in the updated manuscript.

**A. Karpechko (Referee)**
The manuscript has improved compared to the previous version. I think the authors have made a good job. There are only few minor comments which I have on the revised version:

1. I believe the fact that your index does not correlate with the heat flux at 100 hPa (which represent the amount of wave activity entering the stratosphere in extratropics) is important and worth highlighting in conclusions and perhaps also in the abstract. The upper branch of BD is indeed driven by planetary waves and one would expect the 100hPa heat flux being an important proxy for it. But it is likely wave propagation and dissipation within the stratosphere that give raise the variability represented by your index, and this is not captured by the 100hPa heat flux.

Agreed. We think this is interesting and worth pursing in the future.

2. P8L4 : 'It seems surprising that these dynamical signals survive other processes, such as chemistry and radiative effects, on a monthly timescale and we suggest 5 this warrants further investigation, though this is beyond the scope of this paper.'

A dynamical event lasting just for a few days (or a sequence of such events) may influence the monthly mean statistics (temperatures, ozone, etc). So there is no need for an event to last for a month. Perhaps this is a sufficient explanation?

We have removed this sentence from the manuscript.

3. Figure 5 and Section 3.2:

How do you calculate TEMS and EPFD? References to the formulas would be helpful. Also do you use daily mean data or hourly (e.g. 6-hourly) data? Daily mean data may be not sufficient to represent forcing due to some tropical waves. This may be responsible for patchy patterns in the tropics, such as the negative anomalies surrounded by positive anomalies in Fig. 5a. Please give more details about your calculations

We have added the following sentence to the third paragraph of section 3.2: "We used six-hourly model output to calculate the monthly means and use equations 3.5.1 and 3.5.3 from (Andrews and Holton, 1987) to perform the calculations."

*Reference: Andrews, D.G., Holton, J.R. and Leovy, C.B. (1987) Middle Atmosphere, Dynamics. Academic Press Inc., 491 p.*

4. Section 4:

I realize that I do not understand how you construct the index, specifically how you merge the hemispheres. Is it so that June-October values come from SH and November-May from NH? Then why there are several different periods mentioned in Section 4.1?

The DJF and JJA periods are used to find the peak correlation. Once found, we complete the remaining months missing from the index (i.e. MAM and SON) putting N-MA with DJF and M-AM with JJA. We think the confusion may come from the first sentence paragraph 3 in section 4: "For March--May and September--November months, …" so we remove this sentence to reduce confusion.

5. Figure 6: I believe positive values of EP-flux divergence indicate decreased wave activity, which is consistent with weakened meridional circulation and high temperatures in the tropics (high-T composite).

We have corrected this.

6. I think 1-sigma confidence interval is more commonly used wording than 68% confidence interval.

While our confidence intervals are close to Gaussian, we use 68% because it is not necessarily Gaussian in every case.

7. P17L5: 'Nevertheless, use of AR0 or AR1, only appears to influence the uncertainties 5 on the trend estimates in a clear way, and only the mean values to a small degree (see Fig. 13).'

I do not understand this sentence. In any case I think it is enough to consider AR1 results and just skip the AR0 results.

We have removed discussion of AR results; due to an error in our previous analysis we have used AR2 instead since this is actually more appropriate (see response below to Lon Hood).

8. Figure 13 shows that the inclusion of the UBDC index makes the trends more negative, i.e. the trends are increased in the absolute values around 6-10 hPa, although within the uncertainty estimates, am I right? It is not absolutely clear from your text what do you mean by saying that UBDC index leads to a decrease of ozone trends: in terms of absolute values the trends increase.

This correct. We hope the new Figure makes this clearer. The absolute trends are generally positive, but they are almost always reduced in this region by ~0.5-1%; we made a minor adjustment to the text to make this point clear.

9. P18L12: 'temperature and ozone TREND estimates'

We have added this word.

**L. Hood (Referee)**

As stated in my first review, this is overall a valuable and significant effort to improve statistical estimation of stratospheric ozone and temperature trends and natural variability. The presentation is generally excellent and the changes made in the new version of the manuscript are very responsive to the criticisms provided by myself and the other reviewer. However, some minor but important revisions are still needed in my opinion.

First, Figure 13 of the original manuscript was the key figure of the paper because it showed the improvement in trend estimates for the four ozone datasets and the three temperature datasets when the MLSD index was added to the regression model. In the new version of the manuscript, the old Figure 13 has been replaced with a new figure which shows the improvements for only one ozone dataset (SWOOSH) and one temperature dataset (SSU). The improvements are shown for cases with and without including a correction for autocorrelation of the residuals in the regression analysis. This is something of a step backwards because (a) there is no need to include the AR0 results because they do not correct for the autocorrelation; and (b) the improvements for the other three ozone datasets and the other two temperature datasets are not shown. Correction for autocorrelation is standard procedure in regression analyses so only the AR1 results are needed. One could put such a figure in an appendix or in supporting online information if it is desired to show the difference between the AR0 and AR1 results but it should not be in the main paper in my opinion. My suggestion is to correct the old Figure 13 using the AR1 results and put that in the final paper. Then readers can see the improvements for themselves for the various datasets. Adding the AR0 results only adds unnecessary complication to the figure in my opinion.

We have done this as requested, and the original figure has been replaced. We noticed a small error in our analysis routine, which means it was an error to choose AR1. We checked with a Durbin-Watson test and found that while AR1 was indeed at least necessary, AR2 was required to reach an optimal result. We thus have reprocessed all figures and state in the text "We consider the use of `AR2' auto-regressive modelling through the procedure of (Cochrane and Orcutt, 1949) in all cases; see (Tiao et al., 1990) for a discussion of AR. The use of second-order auto-regression was determined after assessing the regression analysis using a Durbin-Watson test, which showed that AR1 was necessary, but not sufficient to account for auto-correlation in the residuals, and that AR2 was sufficient." Note that values (improvements and errors) changed only marginally.

Following this, we have followed Lon's advice in removing discussion on AR0 (or AR1), and replacing Fig 13 with the old version, but with a clearer plot given the multiple extra profiles included.

Second, in the abstract and conclusions, the improvements are stated as maximum improvements ("the index can account for up to 60% of the total variability ... the uncertainty on all multiple linear regression coefficients can be reduced by up to 30% and 25% in temperature and ozone, respectively ..."). While these statements are true, it would be more accurate to give the range of the improvements as a function of pressure level and dataset. Looking at the new Figure 13, the improvement in the

"SWOOSH ozone error bar is indeed 25% at 2.5 hPa but it decreases to about zero by 1 hPa and by 10 hPa. This is a rather strong altitude dependence and readers should be made aware of this."

We have added extra text to both the conclusion and the abstract to address Lon's concern of being clear about where the index is effective. The point regarding

"Also, how do these results depend on the different datasets? Are the improvements less for the other datasets? This should be shown in the final version."

This information is in Figure 10, where the error changes are included in the right panel of each subplot. Nevertheless we now briefly mention the different datasets in the main text, and the improved clarity of the revised Fig 13 makes it much easier to see the error bar improvement.